# Donor-acceptor bulk-heterojunction sensitizer for efficient solid-state infrared-to-visible photon up-conversion

Pengqing Bi[1], Tao Zhang[2], Yuanyuan Guo[3], Jianqiu Wang[2], Xian Wei Chua [1], Zhihao Chen[2], Wei Peng Goh [1], Changyun Jiang [1], Elbert E. M. Chia [3], Jianhui Hou [2] & Le Yang [1,4] ✉

Solid-state infrared-to-visible photon up-conversion is important for spectral-tailoring applications. However, existing up-conversion systems not only suffer from low efficiencies and a need for high excitation intensity, but also exhibit a limited selection of materials and complex fabrication processes. Herein, we propose a sensitizer with a bulk-heterojunction structure, comprising both an energy donor and an energy acceptor, for triplet-triplet annihilation up-conversion devices. The up-conversion occurs through charge separation at the donor-acceptor interface, followed by the formation of charge transfer state between the energy donor and annihilator following the spin statistics. The bulk-heterojunction sensitizer ensures efficient charge generation and low charge recombination. Hence, we achieve a highly efficient solid-state up-conversion device with 2.20% efficiency and low excitation intensity (10 mW cm$^{-2}$) through a one-step solution method. We also demonstrate bright up-conversion devices on highly-flexible large-area substrates. This study introduces a simple and scalable platform strategy for fabricating efficient up-conversion devices.

As an anti-Stokes emission process that converts low-energy photons into high-energy photons, up-conversion (UC) possesses significant application values in sensing, photocatalysis, and photovoltaics, among other fields[1–6]. Compared to liquid-state UC systems, solid-state UC devices, especially solid-state UC from near-infrared (NIR)-to-visible light hold great promise[7–9]. Solid-state UC device is conducive to robust integration into in a wide variety of solar device architectures. Additionally, utilising the superior tissue penetration of NIR light compared to visible light, NIR-to-visible presents significant advantages as an internal light source for various biological applications such bioimaging, drug delivery, photodynamic therapy, and optogenetics[10–13]. Among the various UC mechanisms, triplet-triplet annihilation up-conversion (TTA-

UC) in organic materials, utilizing excitonic states to convert two low-energy photons into one high-energy photon, has attracted substantial attention due to its feasibility with non-coherent photoexcitation[14–17]. Besides, TTA-UC system possesses low excitation power requirements and easily tunable excitation and emission wavelengths[18]. Despite the emergence of novel materials and mechanisms, energy losses within solid-state-based NIR-to-visible TTA-UC systems remain substantial, resulting in low UC efficiencies ($\phi_{UC}$s), which hinders its practical applications[19–21]. Therefore, significant efforts are required to develop high-performance TTA-UC systems with low energy losses.

Developing efficient solid-state NIR-to-visible TTA-UC devices is very challenging because of the complex interplay among various

[1]Institute of Materials Research and Engineering (IMRE), Agency for Science, Technology and Research (A*STAR), 2 Fusionopolis Way, Singapore 138634, Republic of Singapore. [2]State Key Laboratory of Polymer Physics and Chemistry, Institute of Chemistry Chinese Academy of Sciences, Beijing 100190, P. R. China. [3]Division of Physics and Applied Physics, School of Physical and Mathematical Sciences, Nanyang Technological University (NTU), Singapore 637371, Republic of Singapore. [4]Department of Materials Science & Engineering, National University of Singapore (NUS), 9 Engineering Drive 1, Singapore 117575, Republic of Singapore. ✉e-mail: yang_le@imre.a-star.edu.sg

photophysical processes[20,22,23]. For the traditional organometallic sensitizer-based TTA-UC systems, they generally suffered from two severe non-radiative energy losses: exciton annihilation associated with singlet-to-triplet intersystem crossing (ISC) and triplet exciton recombination caused by the sensitizer aggregation[24,25]. In order to reduce the non-radiative energy losses, sensitizers are usually diluted in emitters or other polymer matrices at low concentrations. This strategy significantly weakens the absorption of TTA-UC devices and leads to consistently low $\phi_{UC}s$[26,27]. Moreover, the sensitizers require modification with heavy metal atoms to enhance the spin-coupling effect. However, the heavy metal atoms are often rare or toxic, imposing significant constraints on material selection[28,29]. In addition to the traditional TTA-UC systems mentioned above, other TTA-UC systems can be based on bilayer heterojunction structures. In these bilayer heterojunctions, sensitizer layers are typically fabricated using quantum dots, perovskite, and two-dimensional materials[8,14,30–32]. Emitting layers are deposited on top of these sensitizer layers using either evaporation or solution processing methods. However, there are still some issues with such TTA-UC devices. Firstly, due to the limited diffusion distance of triplet excitons, a significant non-radiative recombination of triplet excitons occurs even when the sensitizer layer is thin[33,34]. Additionally, widely used materials such as lead chalcogenide nanocrystals and lead halide perovskites contain lead (Pb), which are also toxic and environmentally unfriendly. Importantly, these TTA-UC devices also exhibit relatively low performance ($\phi_{UC} < 2.0\%$)[21,35,36]. Another kind of TTA-UC device is based on interface charge recombination, which uses organic small molecules as the sensitizer layer onto which the emitting layer is vacuum-deposited[7]. Using ITIC-Cl as the sensitizer layer and rubrene:DBP blend as the emitting layer, they achieved a TTA-UC device with a $\Phi_{UC}$ of over 2% for NIR-to-visible photon UC. However, in such a device, the singlet excitons generated in the small molecules need to diffuse to the interface between the sensitizer layer and the emitting layer for separation and recombination. Considering the limited diffusion distance of the singlet excitons (tens of nanometers), the sensitizer layer must also be very thin, which sacrifices the photon absorption ability of TTA-UC device, while not being suitable for large-scale production. It should also be noted that, for the bilayer structures, whether composed of hybrid/inorganic materials or organic materials, there exists a limited interface area between the sensitizer and emitter. This limited interface area becomes as a bottleneck where a typical energy transfer process (or charge separation) must occur. Even when the sensitizer and emitter are combined to form a heterojunction structure, the low efficiency of exciton separation still leads to reduced device performance[37]. Therefore, it is necessary to develop solid-state TTA-UC devices with low energy loss, simple device preparation process and versatile material selectivity.

Herein, we propose a strategy of fabricating efficient solid-state TTA-UC devices by using a donor-acceptor bulk-heterojunction (BHJ) as sensitizer. The device can be prepared through a one-step solution method. In our device, the annihilator rubrene are uniformly distributed in the nano interpenetrating networks of PYIT1:PBQx-TCl-based BHJ sensitizer. The energy donor in sensitizer PYIT1 can efficiently absorb NIR photons to generate singlet excitons, which then separate to free charges at the interface between PYIT1 and PBQx-TCl (energy acceptor in sensitizer). Subsequently, holes are transferred from PBQx-TCl to rubrene, forming charge transfer (CT) excitons between PYIT1 and rubrene following the spin statistics (75% triplets and 25% singlets). Finally, two triplets annihilate to form a singlet through TTA on rubrene, leading to UC emission from the rubrene. In this type of device, the non-radiative triplet recombination in traditional TTA-UC device can be significantly reduced. Additionally, there is no limit to the film thickness, which ensures that the device has sufficient photon absorption. As a result, we have achieved a high $\phi_{UC}$ of 2.20% in the PYIT1:PBQx-TCl:rubrene:DBP-based TTA-UC device

with a low threshold power density ($I_{th}$) of 10 mW cm$^{-2}$. We also demonstrate bright TTA-UC devices on highly-flexible large-area substrates using a blade-coating method. This work presents a straightforward and scalable approach for the production of efficient solid-state TTA-UC devices.

## Results
### Materials properties
The molecular structures of PYIT1, PBQx-TCl and rubrene are shown in Fig. 1a. PYIT1 is used here as the energy donor component in the sensitizer due to its decent optoelectronic properties, such as the suitable energy levels and absorption spectrum. Meanwhile, PBQx-TCl acts as the energy acceptor in the sensitizer, facilitating efficient exciton separation and charge transfer. Rubrene is chosen as the annihilator. The normalised absorption spectra of the neat and blend films are shown in Fig. 1b and Supplementary Fig. 1a. The normalised photoluminescence (PL) spectra of the neat films are shown in Supplementary Fig. 1b. The absorption of PYIT1 is mainly in range of 600–900 nm with a main absorption peak centred at 808 nm. For PBQx-TCl, its absorption spectrum is mainly located in the range of 400–600 nm, and it does not absorb in the NIR region. For rubrene, its absorption spectrum is primarily within the range of 400– to 550 nm (Fig. 1c), with two main absorption peaks at 495 and 530 nm. The PL spectrum of rubrene is mainly between 500 and 700 nm, exhibiting two distinct emission peaks at 573 and 605 nm. The absorption spectrum of the PYIT1:PBQx-TCl:rubrene blend film is depicted in Supplementary Fig. 1a. When rubrene is introduced into the PYIT1:PBQx-TCl blend system (PYIT1:PBQx-TCl:rubrene = 1:1:5, weight ratio), an evident change occurs in the relative intensities of PBQx-TCl absorption peaks. The 0-1/0−0 ratio decreases from 1.12 to 1.02. Additionally, a blue shift (-10 nm) is observed in the absorption spectrum corresponding to the PYIT1. The results indicate decreases in the molecular aggregations of both PYIT1 and PBQx-TCl, implying that the introduction of rubrene is distributed within the PYIT1:PBQx-TCl system[38,39]. This distribution is favourable for charge transfer from the PYIT1:PBQx-TCl system to rubrene. Cyclic voltammetry (CV) method has been used to measure the highest occupied molecular orbital (HOMO) and lowest unoccupied molecular orbital (LUMO) energy levels of these components. The CV curves are shown in Supplementary Fig. 1c and the corresponding energy levels are shown in Fig. 1d. The LUMO/HOMO energy levels of PBQx-TCl, PYIT1, and rubrene are −2.70/−5.33, −3.78/−5.68, and −3.23/−5.26 eV, respectively.

For the D-A BHJ sensitizer, a good BHJ is crucial as it enables efficient charge transfer and transport, effectively suppressing geminate and non-geminate recombination. To evaluate the BHJ performance, we initially fabricated organic photovoltaic (OPV) devices, as they are providing a good measure of charge generation and recombination. Previously, we have demonstrated that the PYIT1:PBQx-TCl-based BHJ can be employed to fabricate high-performance OPV cells[40]. Here, we have also prepared fresh PYIT1:PBQx-TCl-based OPV cells for comparison. The current-density-voltage ($J$-$V$) curves and device parameters are shown in Supplementary Fig. 2a and Supplementary Table 1. Under 1-sun illumination (100 mW cm$^{-2}$), the power conversion efficiency (PCE) of the cell is 17.36%, with an open-circuit voltage ($V_{OC}$) of 0.925 V, a short-circuit current density ($J_{SC}$) of 23.89 mA cm$^{-2}$ and a fill factor (FF) of 75.84%. This cell exhibits an external quantum efficiency (EQE) value close to 80% at 808 nm (Supplementary Fig. 2b). Additionally, considering that the TTA-UC devices often operate under monochromatic light, we also evaluated the device performance using an 808-nm laser as the light source. The corresponding $J$-$V$ curves and corresponding device parameters are shown in Supplementary Fig. 2c and Supplementary Table 2. Under the illumination intensities of 10, 47, and 102 mW cm$^{-2}$, we have achieved PCEs of 31.70%, 29.85%, and 27.56%, respectively. The results indicate that under 808 nm excitation, the PYIT1:PBQx-TCl-based BHJ possesses efficient

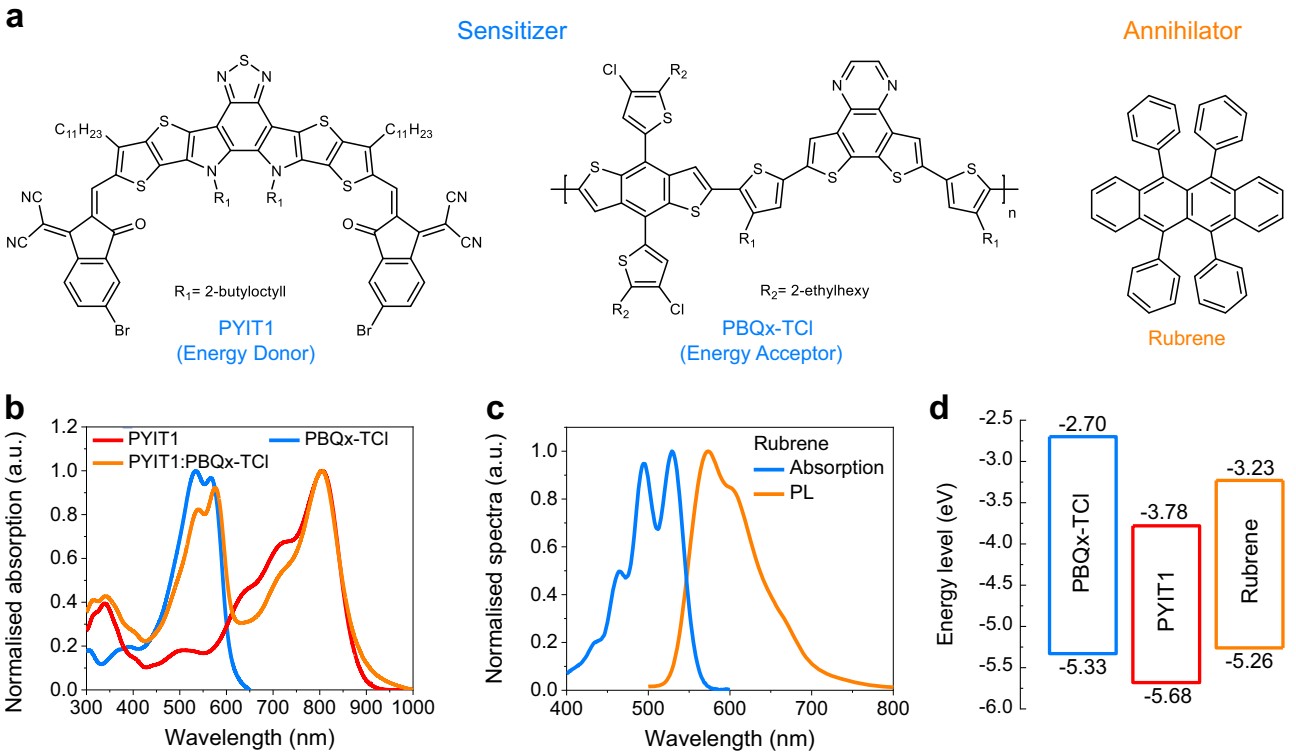

**Fig. 1 | Materials basic properties. a** Molecular structures of PYIT1, PBQx-TCl and rubrene. **b** The normalised absorption spectra of neat PYIT1, neat PBQx-TCl and PYIT1:PBQx-TCl blend films. **c** The normalised absorption and PL spectra of neat rubrene film. **d** The energy level diagrams for PYIT1, PBQx-TCl and rubrene.

photoelectronic conversion capabilities[41]. To further investigate the charge generation upon the introduction of rubrene, we fabricated PYIT1:PBQx-TCl:rubrene-based OPV cells. After introducing rubrene into the active layer blend film, the PCE of the cell decreases to 5.21% (Supplementary Fig. 2a and Supplementary Table 1), but its $V_{OC}$ remains high at 0.902 V, indicating the photogenerated excitons can still be efficiently separated[42]. Comparing the large $V_{OC}$ loss observed in PYIT1:rubrene-based OPV cell (0.64 V) (Supplementary Fig. 3), we reasonably speculate that charge separation primarily occurs between PYIT1 and PBQx-TCl in PYIT1:PBQx-TCl:rubrene BHJ[43]. Additionally, around 800 nm, the cell retains a 50% EQE value. Under the illumination intensities of 10, 50, and 100 mW cm$^{-2}$ of 808 nm laser, the PYIT1:PBQx-TCl:rubrene-based cell can still achieve relatively good PCEs of 16.36%, 9.17%, and 8.19%, respectively (Supplementary Fig. 2d and Supplementary Table 2).

## TTA-UC device performance

The TTA-UC devices were prepared on quartz substrates by spin-coating the blend solutions. As shown in left of Fig. 2a, a bright yellow emission was observed under 808 nm laser illumination (i, Fig. 2a). In contrast, we also placed an additional neat rubrene (ii, Fig. 2a) or PYIT1:PBQx-TCl blend film (iii, Fig. 2a) between the PYIT1:PBQx-TCl:rubrene film and the laser, respectively. When the laser beam passes through these two sets of samples, we can only observe bright luminescence on the PYIT1:PBQx-TCl:rubrene film. There is no UC emission observed in both of the rubrene and PYIT1:PBQx-TCl films. The UC emission spectra are shown in Fig. 2b. The PL peaks of the PYIT1:PBQx-TCl:rubrene film are consistent with that of the neat rubrene film, confirming that the emission originates from rubrene. Compared to the PL spectra of the neat rubrene film, we observed a variation in the relative intensity between the 0−0 and 0-1 peaks in the UC emission spectra. According to previous reports, this can be attributed to the change in the crystalline state of rubrene from the neat film to the blend film[44,45]. $\Phi_{UC}$ is one of the most critical

parameters for evaluating the performance of TTA-UC device. Typically, an integrating sphere is usually used for measuring the absolute photoluminescence quantum efficiency (PLQE) of a material. However, for TTA-UC devices, direct PLQE measured using an integrating sphere can be affected by reabsorption of UC photons by sensitizers[7]. We estimated the reabsorption effect by comparing the UC emission spectra of the PYIT1:PBQx-TCl:rubrene:DBP-based TTA-UC system with (A) and without (B) the integrating sphere (Supplementary Fig. 4a)[46]. The UC emission spectrum obtained in Experiment A represents the spectrum after reabsorption, while the emission spectrum outside the sphere in Experiment B can be considered as the UC emission with minimum reabsorption effect. Comparing the photons obtained from emission spectra A and B (Supplementary Fig. 4b), the effect of reabsorption is particularly severe for our TTA-UC device, resulting in nearly an order of magnitude outcoupling loss. This is due to the significant overlap between our UC emission and the absorption spectra of the D-A sensitizer (Supplementary Fig. 4c). Additionally, as shown in Supplementary Fig. 4d−g, both PYIT1 and PBQx-TCl exhibit relatively large absorption coefficients, at $1.9 \times 10^5$ and $1.0 \times 10^5$ M$^{-1}$ cm$^{-1}$, respectively. Besides, experimental output losses such as wave-guiding, scattering, and inner-filter effects can also induce outcoupling losses. Therefore, a relative method was used to estimate the device $\Phi_{UC}$. In this relative method, a rubrene:DBP film was used as a reference sample. Subsequently, the $\Phi_{UC}$ of TTA-UC devices were measured under different excitation intensities using 808-nm laser. The UC emission spectra of the TTA-UC devices with various excitation intensities are shown in Supplementary Fig. 5a, b. The absorption spectra used for the $\Phi_{UC}$ calculations are shown in Supplementary Fig. 5c, d. As shown in Fig. 2c, the $\Phi_{UC}$ increase with increasing excitation intensity and saturates at high excitation intensity. The saturated $\Phi_{UC}$ for the PYIT1:PBQx-TCl:rubrene device is 0.85%. We also measured the performance of the PYIT1:rubrene TTA-UC device. As shown in Supplementary Fig. 6, the $\Phi_{UC}$ of the PYIT1:rubrene device is 0.09%.

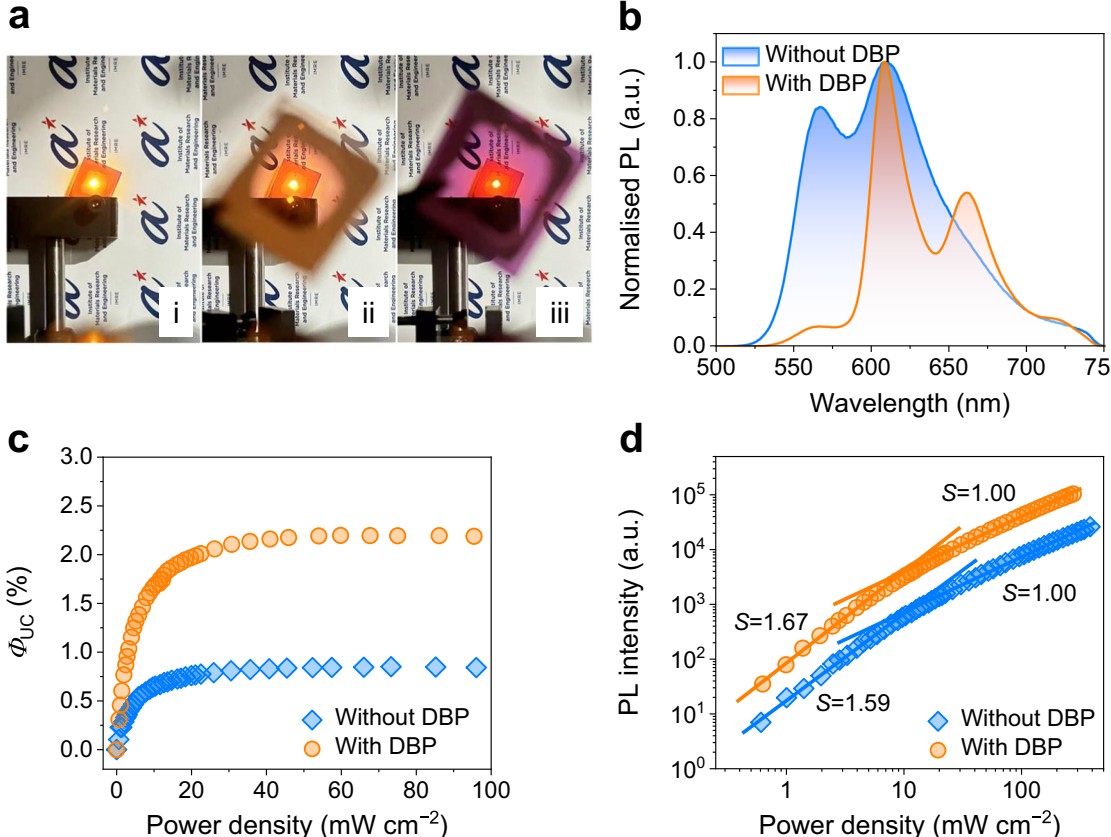

**Fig. 2 | TTA-UC device performance. a** Photographs of the PYIT1:PBQx-TCl:rubrene film alone (i), and with an additional neat rubrene film (ii) or PYIT1:PBQx-TCl film (iii) between the laser and the PYIT1:PBQx-TCl:rubrene film, under 808-nm laser illuminations. **b** UC emissions from the PYIT1:PBQx-TCl:rubrene and PYIT1:PBQx-TCl:rubrene:DBP films under 808-nm laser illuminations. **c** The $\Phi_{UC}$ of the PYIT1:PBQx-TCl:rubrene and PYIT1:PBQx-TCl:rubrene:DBP UC devices as a function of power density. **d** The UC emission intensities of the PYIT1:PBQx-TCl:rubrene and PYIT1:PBQx-TCl:rubrene:DBP UC devices as a function of power density. S represents slope.

Due to the energetics of the triplet state, which is approximately half the energy of the singlet state, rubrene can engage in both TTA and singlet fission (SF). Consequently, tetraphenyldibenzoperiflanthene (DBP) is often introduced into the rubrene layer to efficiently capture singlet states from rubrene, thereby suppressing SF and enhancing the corresponding device performance[47]. The molecular structure, absorption and PL spectra of DBP are shown in Supplementary Fig. 7. Here, we introduce DBP into the PYIT1:PBQx-TCl:rubrene system as the emitter with a weight ratio relative to rubrene of 0.75 wt% and fabricated the PYIT1:PBQx-TCl:rubrene:DBP UC device using the same one-step method. The main emission peak of the DBP-doped device is located at 608 nm (Fig. 2b). The saturated $\Phi_{UC}$ of the DBP-doped UC device is 2.20%, which is two times higher than that of the PBQx-TCl:PYIT1:rubrene device (Fig. 2c). It should be noted that the $\Phi_{UC}$ results obtained using relative methods can be sensitive to the condition of the optical setup, leading to variations in the reported values[48]. Therefore, it is difficult to directly compare results from different reports. However, we believe that the TTA-UC systems reported here should be considered among the highest-performing TTA-UC systems currently available. As widely recognised, the UC emission through the TTA process exhibits a distinct dependence on the incident power density, transitioning from quadratic to linear as excitation power increases. This transition is attributed to the alteration of the dominant triplet decay pathway in annihilators, characterised by primary geminate triplet recombination at low excitation power and non-geminate triplet recombination at high excitation power[49]. We further explore the UC emission intensity as a function of incident power density. The UC emission intensity as a function of the incident power density is shown in Fig. 2d on a dual-logarithm scale. The fitted slopes of the PYIT1:PBQx-TCl:rubrene and PYIT1:PBQx-TCl:rubrene:DBP UC intensities at lower excitation intensities are 1.59 and 1.67, respectively, and these became 1.00 at higher intensities, which further confirm the UC emissions are coming from the TTA mechanism. The transition point, $I_{th}$, is another important parameter[14]. For traditional TTA-UC systems, $I_{th}$ can be expressed by the equation $I_{th} = (\alpha \oslash_{TET} k_{TTA} \tau_{TE}^2)^{-1}$[50]. The PYIT1 possesses a high molar extinction coefficient of $1.9 \times 10^5$ $M^{-1}$ $cm^{-1}$. Furthermore, compared to the traditional TTA-UC systems that rely on ISC and energy transfer processes, the working mechanism of our device is based on efficient charge generation and recombination processes. This significantly reduces the non-radiative energy losses associated with triplet diffusion and energy transfer processes in traditional TTA-UC devices. This ensures the accumulation of a high density of triplet excitons within the rubrene domains even under low-energy excitation. Additionally, we also measured the lifetime of the UC emission, which was found to be 22 µs (Supplementary Fig. 8). These highly efficient processes enable the achievement of low threshold power densities for our PYIT1:PBQx-TCl:rubrene and PYIT1:PBQx-TCl:rubrene:DBP UC devices, with $I_{th}$ values of 8 and 10 mW cm$^{-2}$, respectively. These $I_{th}$ values are among the lowest reported to date, indicating a very low threshold power needed to elicit efficient UC.

## Morphological and crystalline properties

The film formation processes and the distribution of components are crucial to the one-step method prepared TTA-UC devices. Here, various morphology characterisation techniques were employed to

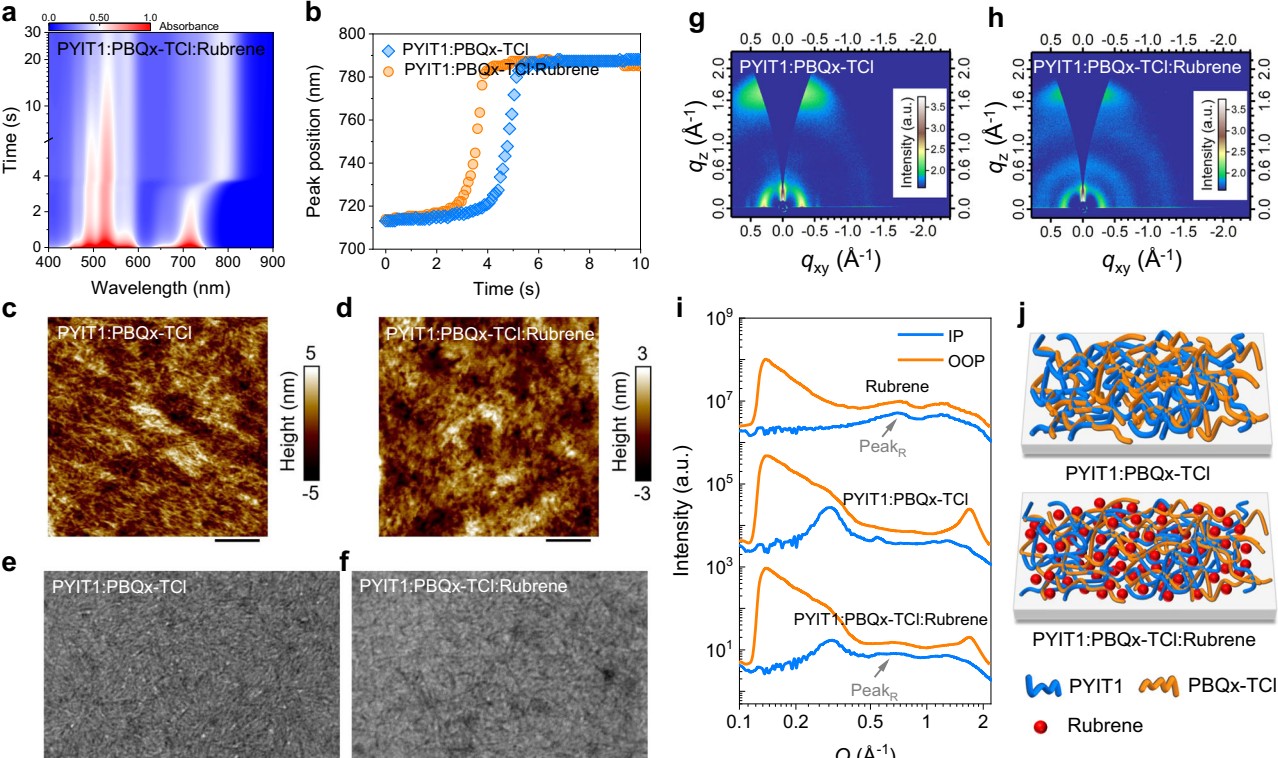

**Fig. 3 | Morphological and crystalline properties. a** Time-dependent contour maps of absorption spectra for the PYIT1:PBQx-TCl:rubrene system. **b** Time evolution of the PYIT1 absorption peak locations in PYIT1:PBQx-TCl and PYIT1:PBQx-TCl:rubrene systems. **c**, **d** AFM height images of the PYIT1:PBQx-TCl and PYIT1:PBQx-TCl:rubrene films. The scale bar is 400 nm. **e**, **f** TEM images of the PYIT1:PBQx-TCl and PYIT1:PBQx-TCl:rubrene films. The scale bar is 200 nm. **g**, **h** 2D GIWAXS patterns of the PYIT1:PBQx-TCl and PYIT1:PBQx-TCl:rubrene films. The colour scales represent the log of diffraction intensity, in the unit of counts. **i** The 1D GIWAXS integration curves along IP and OOP directions. Peak$_R$ represents the diffraction peak of rubrene. **j** Schematic illustration of the PYIT1:PBQx-TCl:rubrene film morphology.

investigate the morphological and crystalline properties. In-situ absorption measurements were conducted to study the phase transition processes from solution to film. The temporal evolution of the absorption images and spectra of the neat and blend systems are shown in Fig. 3a, Supplementary Figs. 9 and 10. The phase transition process of PYIT1 can be revealed by plotting the location of the characteristic absorption peak (0–0 peak) as a function of time. As illustrated in Supplementary Fig. 10a, PYIT1 exhibits four distinct phase transition processes: solvent evaporation (0-1 s), diffusion and nucleation (1–1.5 s), crystal growth (1.5-1.7 s), and the dried film stage (after 1.7 s)[51]. The neat rubrene system does not show an obvious multiphase transition process. Besides, there is no obvious change in the absorption peak positions in the liquid phase and solid phase (~525–~529 nm), as shown in Supplementary Fig. 10b, indicating that rubrene has strong solution aggregation effect. In the PYIT1:PBQx-TCl system, PYIT1 also exhibits four phase transition processes (Fig. 3b). In detail, it undergoes a rapid solvent evaporation stage from 0 to 3.4 s, then enters the nucleation stage (3.4-4.5 s), followed by crystal growth (4.5–5.3 s), and finally reaches the dried film stage at 5.3 s. However, in comparison, PYIT1 in the PYIT1:PBQx-TCl:rubrene system exhibits a relatively short crystallization process, including the solvent evaporation stage (0–2.9 s), the nucleation stage (2.9–3.5 s), rapid crystal growth (3.5-3.9 s), and the dried film stage (after 3.9 s), resulting in smaller crystal sizes. Additionally, in the solid state, the absorption peak of PYIT1 shows an obvious blue-shift after introducing rubrene, indicating that the molecular packing order between PYIT1 molecules in the PYIT1:PBQx-TCl system is stronger than that in the PYIT1:PBQx-TCl:rubrene system[52]. For PBQx-TCl, due to its strong solution aggregation effect, it is difficult to investigate its phase transition process by tracing its peak position changes. Here, the evolution of the absorption

peak (0–0) intensity over time is used to study its phase transition process. In the PYIT1:PBQx-TCl system, PBQx-TCl also shows four distinct processes (Supplementary Fig. 10c). However, after the introduction of rubrene, PBQx-TCl exhibits only two processes, with the absence of a nucleation growth stage, suggesting that the presence of rubrene also suppresses the crystalline size of PBQx-TCl. The phase transition processes of rubrene in the blend system exhibit similar trends with that of the neat system (Supplementary Fig. 10d). However, there is an obvious blue shift in the absorption peak, which suggests a significant decrease in crystallinity of rubrene in the blend film.

We use atomic force microscopy (AFM) to analyse the surface morphology of the thin films (Fig. 3c, d, and Supplementary Fig. 11). PYIT1:PBQx-TCl film exhibits a smooth surface with a root mean square roughness ($R_q$) of 1.43 nm. In comparison, the PYIT1:PBQx-TCl:rubrene film shows smaller roughness ($R_q$ = 0.93 nm), which is attributed to the uniform distribution of rubrene in the PYIT1:PBQx-TCl system, reducing the aggregations of PYIT1 and PBQx-TCl. In the AFM phase images, the coarser fibril morphology can be clearly seen in the PYIT1:PBQx-TCl film, suggesting the formation of large-sized fibril-like structures by PYIT1 and PBQx-TCl. After adding rubrene, the fibril size of PYIT1:PBQx-TCl decreases, and rubrene does not show large aggregation, but is evenly distributed in the fibril network morphology. Furthermore, we used transmission electron microscope (TEM) to study the internal morphology of the films (Fig. 3e, f). In the PYIT1:PBQx-TCl film, we observe larger-sized and more densely distributed fibril-like structures, consistent with the AFM phase images. In the PYIT1:PBQx-TCl:rubrene system, the fibre size decreases, and the distribution becomes sparse. Subsequently, the crystalline properties of the neat and blend films were studied by grazing incidence wide-angle X-ray scattering (GIWAXS). The 2D patterns and corresponding

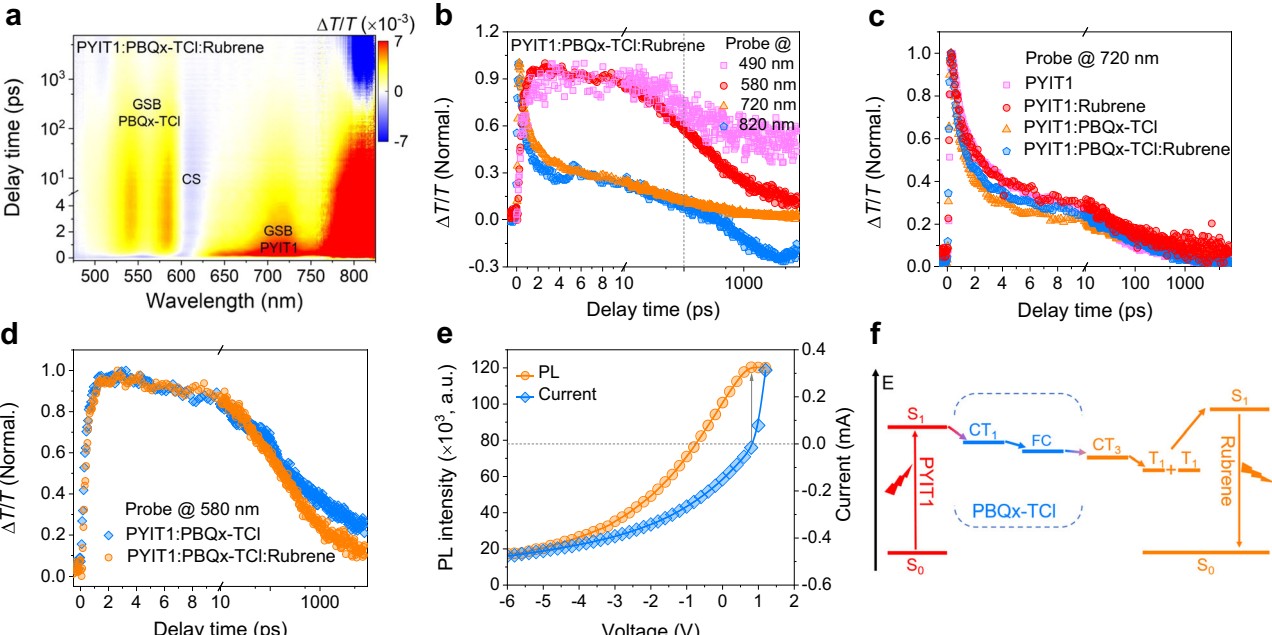

**Fig. 4 | The TTA-UC device working mechanism. a** The TA spectral map of PYIT1:PBQx-TCl:rubrene film. **b** The decay curves at various wavelengths of PYIT1:PBQx-TCl:rubrene films. **c** The GSB dynamics of PYIT1 traced at 720 nm in both neat and blend films. **d** The hole transfer process in PYIT1:PBQx-TCl and PYIT1:PBQx-TCl:rubrene films. **e** Dependence of the simultaneously measured UC emission intensity and photocurrent of the PYIT1:PBQx-TCl:rubrene OPV cell. **f** The working mechanism schematic diagram of TTA-UC device. $S_0$, $S_1$, $T_1$ and FC represent ground state, first excited state and first triplet excited state and free carrier, respectively.

1D profiles of PYIT1, PBQx-TCl and rubrene are shown in Supplementary Fig. 12a–d. For PYIT1, only the out-of-plane (OOP) direction exhibits distinct π-π stacking peaks, suggesting a face-on dominated molecular packing orientation (Supplementary Fig. 12a)[53]. For PBQx-TCl, clear scattering peaks are observed in both the in-plane (IP) direction (010 π-π stacking peak) and OOP direction (100, 200, and 300 peaks), indicating a face-on and edge-on coexistence of molecular packing orientations (Supplementary Fig. 12b). The neat rubrene film exhibits sharp scattering peaks accompanied by many diffraction spots, which indicates rubrene forms randomly oriented crystals (Supplementary Fig. 12c)[54]. The 2D patterns and corresponding 1D profiles of PYIT1:PBQx-TCl, PYIT1:PBQx-TCl:rubrene films are shown in Fig. 3g-i. The PYIT1:PBQx-TCl film exhibits a clear edge-on molecular packing orientation with the π-π stacking peak at 1.69 Å⁻¹. According to the Scherrer formula, the molecular packing distance ($d$) and coherence length (CL) are 3.72 and 28.55 Å (Supplementary Fig. 12e), respectively[55]. After introducing rubrene, a reduction in the intensity of the π-π stacking peak is observed, and the CL decreases to 22.43 Å (Supplementary Fig. 12f). Additionally, rubrene in the blend film only shows broad peaks at $q \approx 0.68$ Å⁻¹ and $q \approx 1.30$ Å⁻¹ with isotropy (Fig. 3h), also confirming a significant reduction in the crystallinity of rubrene. This is consistent with the transition of rubrene from a crystalline state to an amorphous state as reported in literature[56]. Therefore, rubrene exists in an amorphous state within the blend film and is uniformly distributed within the fibrous network formed by PYIT1 and PBQx-TCl. A schematic illustration of the PYIT1:PBQx-TCl:rubrene film morphology is shown in Fig. 3j. This morphology is favourable for rubrene to receive free charge carriers and undergo triplet exciton generation.

## TTA-UC device working mechanism

We investigate the UC working mechanism using transient absorption (TA) spectroscopy. An 800 nm wavelength was used to selectively excite the PYIT1. For TA measurements, employing high excitation powers may induce two-photon absorption (TPA) effect in rubrene. To avoid this phenomenon, we used excitation powers <10 μJ cm⁻². As shown in Supplementary Fig. 13, under this excitation power, rubrene does not exhibit TPA effect. The TA images of PYIT1, PYIT1:rubrene, PYIT1:PBQx-TCl and PYIT1:PBQx-TCl:rubrene films and the decay curves at various wavelengths are shown in Fig. 4a, b and Supplementary Fig. 14. The neat PYIT1 film shows obvious ground state bleaching (GSB) signal in the range of 650–800 nm (Supplementary Fig. 14a). For the binary PYIT1:rubrene-only film, the GSB signal of PYIT1 is also exhibited (Supplementary Fig. 14b), but as the GSB signal of PYIT1 quenches, a weak GSB signal around 490 nm, corresponding to rubrene, increases (Supplementary Fig. 14e). This indicates that only a small fraction of the holes in PYIT1 can be directly transferred to rubrene. Furthermore, the GSB signal of PYIT1 in the PYIT1:rubrene film decays only slightly faster compared to the GSB signal in the PYIT1 neat film (Fig. 4c and Supplementary Fig. 4d), also indicating relatively weak hole transfer from PYIT1 to rubrene[57]. The weak hole transfer from PYIT1 to rubrene should be the main reason for the low $\Phi_{UC}$ of PYIT1:rubrene TTA-UC device. In addition, an absorption signal at ~820 nm can be observed, which began to appear at ~10 ps (Supplementary Fig. 14b). We classify it as the charge transfer intermediate state associated with PYIT1 and rubrene. In the PYIT1:PBQx-TCl film, strong GSB signals related to PBQx-TCl are observed in the range of 550–650 nm (Supplementary Fig. 14c), indicating efficient hole transfer from PYIT1 to PBQx-TCl. Besides, no absorption signal is observed around 820 nm in the PYIT1:PBQx-TCl system, similarly suggesting that the signal at 820 nm mentioned above is related to the charge transfer intermediate state between PYIT1 and rubrene[58]. After introducing rubrene into the PYIT1:PBQx-TCl film, obvious PBQx-TCl GSB signals are also observed (Fig. 4a). In addition, in PYIT1:PBQx-TCl and PYIT1:PBQx-TCl:rubrene films, the rising edge of PBQx-TCl GSB signal has no significant difference (Fig. 4d), indicating that holes still primarily transfer to PBQx-TCl. After reaching the maximum value, the PBQx-TCl GSB signal in the PYIT1:PBQx-TCl:rubrene film quenches much faster than in the PYIT1:PBQx-TCl film, which suggests that the holes subsequently transfer from PBQx-TCl to rubrene. For electrons,

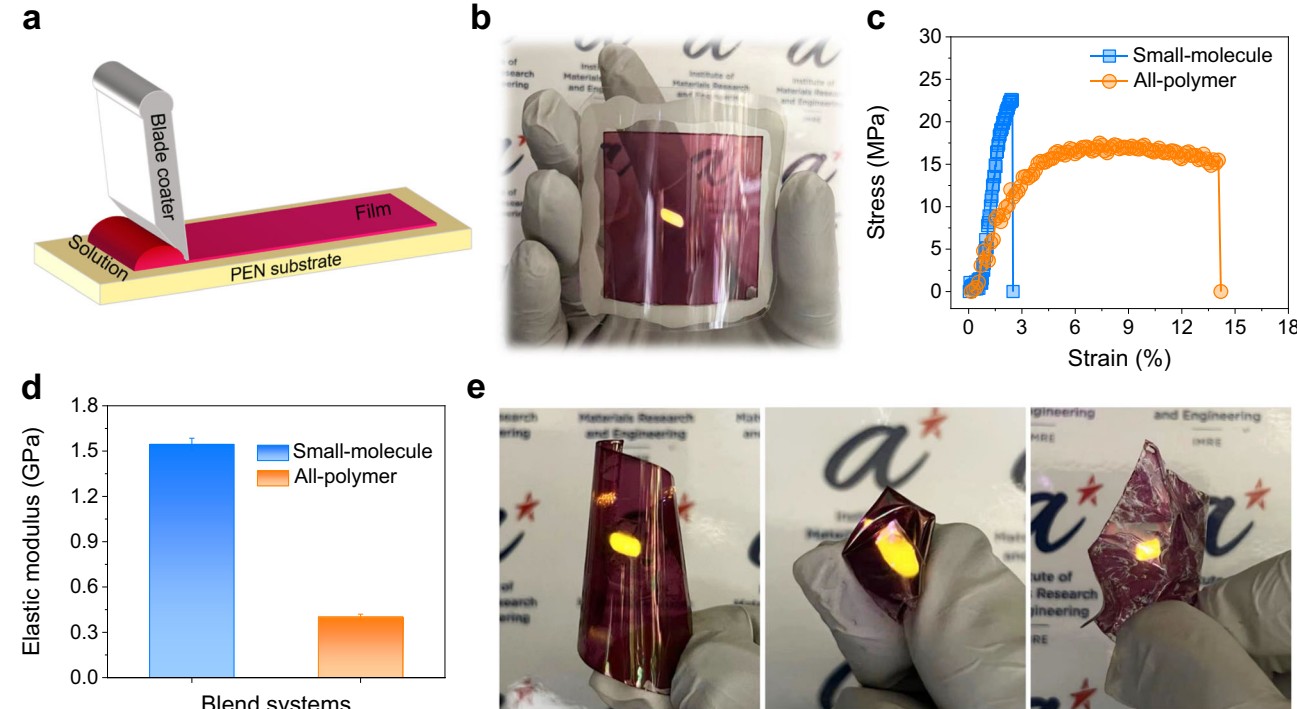

**Fig. 5 | Flexible and large-area TTA-UC devices. a** Schematic diagram of the blade-coating method. **b** Photograph of the 25 cm² large-area TTA-UC device prepared on a PEN substrate. **c** The stress-strain curves of PYIT1:PBQx-TCl:rubrene and PYIT:PBQx-TCl:rubrene films. **d** The elastic moduli of PYIT1:PBQx-TCl:rubrene and PYIT:PBQx-TCl:rubrene films. Error bars are one statistical standard deviation. **e** Photograph of the TTA-UC devices prepared on PI substrates.

due to energy level constraints, they can only remain in the LUMO of PYIT1. Importantly, in the PYIT1:PBQx-TCl:rubrene film, we also observe the evident absorption signal at ~820 nm, which begins to appear at about 40 ps (Fig. 4b). The appearance of this signal indicates that electrons and holes recombine at the interface of PYIT1 and rubrene, generating charge transfer excitons following the spin statistics. The charge transfer triplet excitons can be subsequently transferred to the triplet states of rubrene via energy transfer. Furthermore, we also observe a photoinduced absorption (PIA) signal around 510 nm in the PYIT1:PBQx-TCl:rubrene system, as shown in Supplementary Fig. 15a–c. In contrast, such signal was not observed in the PYIT1:PBQx-TCl system. Similar PIA features in the same spectral region have been explicitly assigned to $T_1 \rightarrow T_n$ transitions in rubrene[59]. However, it is difficult to accurately extract this absorption signal separately due to overlap with the GSB signals of PYIT1. To further confirm that the PIA signals originate from the triplet state of rubrene, TA spectroscopy was further carried out using an EOS nano- and microsecond (ns-μs) spectrometer. As shown in Supplementary Fig. 15d, e, these PIA peaks around 510 nm align well with the previously observed absorption peaks of the rubrene triplet state[59]. Moreover, the PIA signal probed at 515 nm shows a μs-scale lifetime (Supplementary Fig. 15f). These results confirm the formation of the generation of rubrene triplet states. The generation of the triplet states is then followed by the fusion of the triplet states of two rubrene molecules into a higher-lying, emissive singlet exciton, thus completing the UC emission process.

The TA spectra in the NIR region with 800 nm excitation were also measured to demonstrate the generation of CT states between PYIT1 and PBQx-TCl, and the results are shown in Supplementary Fig. 16. For PYIT1:PBQx-TCl, we observed distinct absorption of the PYIT1 localized exciton (LE) at 850 nm, and at 940 nm, absorption corresponding to the CT exciton between PYIT1 and PBQx-TCl is also observed[60]. The decay kinetics of 850 nm and 940 nm signals also exhibit characteristics consistent with those of LE and CT excitons. For the PYIT1:PBQx-

TCl:rubrene system, we observed absorption signals corresponding to the LE of PYIT1 and the CT exciton between PYIT1 and PBQx-TCl at the similar positions. Additionally, in both of the PYIT1:PBQx-TCl and PYIT1:PBQx-TCl:rubrene TA spectra in visible region, we also extracted absorption signals at 615 nm. According to previous report, these signals correspond to charge-separated (CS) states between PYIT1 and PBQx-TCl[60]. The decay dynamics of the CS states are shown in Supplementary Fig. 17. The results further confirm our description that photogenerated excitons initially separated at the PYIT1:PBQx-TCl interface, followed by subsequent TTA-UC processes.

We further validate the working mechanism using electrical measurements. We prepared OPV cells with PYIT1:PBQx-TCl and PYIT1:PBQx-TCl:rubrene blends as active layers, and measured their corresponding high-sensitivity EQE (HEQE) and electroluminescence (EL) spectra. As shown in Supplementary Fig. 18, the optical bandgaps ($E_g$) of PYIT1:PBQx-TCl- and PYIT1:PBQx-TCl:rubrene-based cells are 1.47 and 1.48 eV, respectively. Through fitting the HEQE and EL spectra, we find that both systems have an identical charge transfer state energy ($E_{CT}$) value of 1.45 eV. Since the HOMO energy level of rubrene is higher than that of PBQx-TCl, the results indicate that charge transfer states are formed at the interface of PYIT1 and PBQx-TCl in both systems[61]. Additionally, we observe very similar EL spectra in both systems, further confirming the similarity in the positions of their CT states. The CT excitons can be separated into free charges at the PYIT1 and PBQx-TCl interface. We then further investigate the UC emission under different applied voltages in the PYIT1:PBQx-TCl:rubrene-based OPV cell to further demonstrate that the UC emission originates from the free charges generated as mentioned above. The measured PL spectra are shown in Supplementary Fig. 19a. Under irradiation with an 808-nm laser, the UC emission intensity and current as a function of applied voltage are shown in Supplementary Fig. 19b and Fig. 4e. When a high reverse voltage is applied (< −3 V), resulting in a stronger external electric field, most of the generated free charges are extracted by the external circuit to form a loop current, which leads to the

maximum current intensity. Consequently, the UC emission is very weak, which is due to the limited number of charge carriers recombining at the interface between PYIT1 and rubrene. As the applied voltage gradually weakens, the external electric field intensity gradually decreases. Therefore, the ability of the external circuit to extract free charges decreases, and the current gradually decreases. At this time, the probability of photogenerated free carriers recombining at PYIT1 and rubrene increases, resulting in a gradual increase in UC emission intensity. When the applied voltage increases to the $V_{OC}$, this condition is the same as that of the BHJ ternary-blend film without electrodes, in which the free charges are mainly recombined at the interface of PYIT1 and rubrene. We can observe that the UC emission reaches its maximum value. As the voltage is further increased, the UC intensity slightly decreases because external injected charges recombine with the photogenerated charges. Furthermore, we observed that this cell operates as an OLED with a threshold turn-on voltage ($V_{th}$) of 1.2 V, as shown in Supplementary Fig. 19c. When it exceeds 1.2 V, it will affect our assessment of UC intensity in the OPV measurement. This TTA-UC processes are summarised in a schematic diagram in Fig. 4f.

## Mechanical properties and large-area flexible devices

Large-area roll-to-roll printing is one of the advantages of solution-processed devices, and this processing method requires the use of flexible substrates[62]. To investigate the application of such device on flexible substrate, we use blade-coating to fabricate a TTA-UC device on a 5 × 5 cm² flexible polyethylenenaphthalate (PEN) substrate. Schematic diagrams of blade-coating method and the fabricated device photos are shown in Fig. 5a, 5b. Bright yellow UC emission can be observed under the excitation of 808-nm laser illumination. The mechanical properties of the UC films were also explored using film-on-elastomer (FOE) measurements. The histogram of crack-onset strain (COS) for the films is displayed in Fig. 5c. The fracture strain and elastic moduli are determined to be 2.50% and 1.54 GPa for the PYIT1:PBQx-TCl:rubrene film. However, these values are not sufficient to meet the requirements of wearable devices[63]. It is well-studied that the mechanical properties of all-polymer films are much better than those of small-molecule films[64]. Here, PY-IT with a molecular weight of 20,498 was further used to fabricate a TTA-UC device with improved mechanical properties. The mechanical stability of PYIT:PBQx-TCl:rubrene is significantly enhanced with a larger fracture strain and elastic moduli of 14.2% and 0.40 GPa, respectively (Fig. 5d). The improvement is mainly attributed to the better interlacing and entanglement of polymer chains[65]. With the improved mechanical stability, we further prepared a highly flexible UC device on a polyimide (PI) substrate (Fig. 5e). Under the 808-nm laser illumination, bright UC emission also can be seen. Even after numerous folds and crumpling, the UC device maintains its bright UC emission. These results demonstrate that such TTA-UC device fabrication method can be easily scaled up on flexible substrates.

## Discussion

In summary, an efficient solid-state NIR-to-visible TTA-UC device with a simple fabrication process was demonstrated. By mixing the donor-acceptor (PYIT1:PBQx-TCl) BHJ type sensitizer with annihilator rubrene, the high-performance TTA-UC device can be prepared using a one-step, all-solution-based processing method. Morphological characterisations reveal that rubrene are uniformly distributed within the fibrous network morphology formed by PYIT1 and PBQx-TCl. The working mechanism involves optical excitation generating excitons in the energy donor PYIT1, followed by exciton separation at the PYIT1:PBQx-TCl interface. The holes are subsequently transferred to rubrene via PBQx-TCl, where triplet charge transfer states are formed at the interface between PYIT1 and rubrene, followed by the generation of triplet excitons of rubrene. These triplet excitons in rubrene undergo TTA to generate singlet excitons, which eventually emit. Based on the PYIT1:PBQx-TCl:rubrene system, we have achieved an UC efficiency of 0.85% with an $I_{th}$ of

8 mW cm$^{-2}$. By introducing an appropriate amount of emitter DBP, the UC efficiency can be further enhanced to 2.20% ($I_{th}$ = 10 mW cm$^{-2}$). This is one of the highest efficiencies for solid-state NIR-to-visible UC devices. Furthermore, we have demonstrated the fabrication of large-area TTA-UC devices on a flexible PEN substrate using the blade-coating method. Additionally, by replacing the sensitizer with a polymer PY-IT, we can significantly improve the mechanical stability of these devices, which enable the fabrication of ultra-flexible TTA-UC devices on PI substrates. Our findings offer pathways for the preparation of high-efficiency, large-area, flexible solid-state NIR-to-visible UC devices for various practical applications.

## Methods

### Reagents and materials
PEDOT:PSS (Clevios P VP Al 4083) was purchased from H.C. Starck Co., Ltd. PYIT1, PBQx-TCl, and PY-IT were purchased from Solarmer Materials, Inc. Rubrene and DBP were purchased from Luminescence Technology Corp. All the chemicals were used as received, without further purification.

### Steady-state characterisations
A Shimadzu Spectrophotometer UV-3600 and a CHI650D electro-chemical workstation were used to collect the UV-vis absorption spectra and the CV curves, respectively. In-situ UV-vis absorption spectra were measured by using an OCEAN-FX-VIS-NIR-ES spectro-meter and a light source of HL-2000-FHSA (Ocean Optics, Inc.) The absorption spectra were calculated from the transmission spectra using the following Eq. 1:

$$A_\lambda = -\log_{10}(T) \tag{1}$$

where $A_\lambda$ is the absorption intensity at the wavelength of $\lambda$. $T$ is transmittance.

### Morphological characterisations
A Bruker Nanoscope V AF microscope and a JEOL JEM-2100PLUS electron microscope were used to measure the AFM and TEM images, respectively. GIWAXS measurements were performed using an XEUSS SAXS/WAXS system (XENOCS, France).

### TA measurements
An Ultrafast Helios pump-probe system and a regenerative amplified laser system from Coherent were used to perform the TA measurements. An 800 nm pulse with a repetition rate of 1 kHz and a length of 35 fs, generated by a Ti:sapphire amplifier, was used as the excitation light. The time delay between the pump and probe was controlled using a motorised optical delay line with a maximum delay time of about 8 ns. The nano- and microsecond TA spectra were measured using a pump-probe nanosecond TA spectrometer with an extended time window (EOS, Ultrafast Systems).

### OPV cells fabrication procedures
The patterned ITO substrates were consecutively cleaned in detergent, de-ionised water, acetone, and ethanol for 20 min, respectively. After UV-ozone treatment for 20 min, PEDOT:PSS solution was spin-coated on the ITO substrate to form a hole transporting layer (~15 nm). The PEDOT:PSS layer was then annealed at 150 °C for 15 min in air condition. After which, the optimised blend solutions of PYIT1:PBQx-TCl (1:1, weight ratio) PYIT1:PBQx-TCl:rubrene (1:1:5, weight ratio) in toluene (6 mg mL$^{-1}$ for PBQx-TCl) were then spin coated on the ITO/PEDOT:PSS layers for binary and ternary OPV cells, respectively. The active layers need to be annealed at 100 °C for 10 min. After which, PDINN layers with ~5 nm thicknesses were spin coated on the active layers. Finally, ~150 nm Ag layers were thermally evaporated through shadow masks.

## OPV cells performance measurements

The *J-V* curves of the OPV cells under AM 1.5 G illumination were measured with a Keithley 2400 and a solar simulator (XES-70S1, SAN-EI Electric Co., Ltd.). A certified standard silicon solar cell (SRC-2020, 174 Enlitech) was used to calibrate the light intensity. An integrated IPCE measurement system (QE-R3011, Enli Technology Co. Ltd.) was used to measure the EQE spectra. For the device performance under 808-nm laser illumination, the laser power intensities were measured by a power meter (S142C, Thorlabs).

## HEQE and EL spectra fitting

$E_g$ is the energy at the point of intersection between HEQE and EL spectra. $E_{CT}$ is the energy at the point of intersection between CT absorption and emission, which can be obtained through fitting the HEQE and EL spectra with Eqs. 2, 3 and 4.

$$\text{EQE}_{PV,CT}(E) = \frac{f}{E\sqrt{4\pi\lambda k_B T}} \exp\left(\frac{-(E_{CT}+\lambda-E)^2}{4\lambda k_B T}\right) \quad (2)$$

$$\text{EQE}_{EL,CT}(E) = E\frac{f}{\sqrt{4\pi\lambda k_B T}} \exp\left(\frac{-(E_{CT}+\lambda-E)^2}{4\lambda k_B T}\right) \quad (3)$$

$$\text{EQE}_{PV}(E) \propto \text{EL}(E)E^{-2}\exp\left(\frac{E}{k_B T}\right) \quad (4)$$

where $k_B$ is Boltzmann's constant. $E$ is the photon energy, and $T$ is the absolute temperature. The reorganisation energy, $\lambda$, and a measure of the strength of the donor-acceptor coupling, $f$, also can be obtained by fitting the HEQE and EL spectra.

## UC device fabrication and performance characterisations

The procedures for cleaning the TTA-UC device substrates are the same as for OPV devices. The PYIT1:PBQx-TCl:rubrene (1:1:5, weight ratio) or PYIT1:PBQx-TCl:rubrene:DBP (1:1:5:0.0075, weight ratio) blend solution, with a PYIT1 concentration of 6 mg mL$^{-1}$, was spin-coated onto the substrate in a glovebox at a spinning speed of 2000 rpm. Subsequently, the fabricated film was annealed at 100 °C for 10 min. Before measuring, the fabricated TTA-UC devices were encapsulated in a glovebox to isolate it from moisture and oxygen. For the relative method to calculate the performance of TTA-UC devices, the PL intensity of a rubrene:DBP blend film was used as a reference. Here, the maximum $\Phi_{UC}$ of the UC device is theoretically 50%. The absolute PLQE of the rubrene:DBP film under an excitation wavelength of 488 nm, measured by the spectrometer system with the integration sphere is 48.98 ± 2.02%. The EQE of the UC emission can be calculated by the following Eq. 5:

$$\text{EQE} = \left(\frac{I_{UC}}{I_{std}}\right)\left(\frac{P_{std}}{P_{UC}}\right)\left(1-10^{-A_{std}}\right)\text{QE}_{std} \quad (5)$$

where $A$ is the absorbance at the excitation wavelength. $I$ is the PL intensity and $P$ is the number of photons irradiated from the excitation light source. Subscripts UC and std refer to the UC emission and the standard sample, respectively. The $\Phi_{UC}$ is defined as the ratio of the number of upconverted photons emitted to the number of low-energy photons absorbed, which can be further expressed by Eq. 6[24].

$$\Phi_{UC} = \text{EQE}/(1-10^{-A_{UC}}) \quad (6)$$

Then, the $\Phi_{UC}$ can be calculated through Eq. 7.

$$\Phi_{UC} = \left(\frac{I_{UC}}{I_{std}}\right)\left(\frac{P_{std}}{P_{UC}}\right)\left(\frac{1-10^{-A_{std}}}{1-10^{-A_{UC}}}\right)\text{QE}_{std} \quad (7)$$

## Reporting summary

Further information on research design is available in the Nature Research Reporting Summary linked to this article.

## Data availability

Source data are provided with this paper. All other data generated or analyzed during this study are included in the published article and its Supplementary Information. Source data are provided with this paper.

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

## Acknowledgements

This research is supported by the Singapore National Research Foundation Fellowships (NRFF) (NRF-NRFF15-2023-0011) (L. Y.); the A*STAR Central Research Fund (L. Y.); the Young Individual Research Grant (YIRG) (OUNI230401dENTIRG) (L.Y.); A*STAR Graduate Academy studentships (X. W. C., L. Y.); and the Singapore Ministry of Education Academic Research Fund Tier 3 (MOE2018-T3-1-002) (Y. G., E. E. M. C.).

## Author contributions

P.B. and L.Y. conceived and directed this project. P.B. fabricated and characterised the TTA-UC devices. T.Z. and J.W. conducted the AFM, TEM measurements and OPV device fabrication and characterization. Y.G. and E.C. provided TA results and corresponding analysis. Z.C. carried out the in-situ absorption measurement. P.B. wrote the manuscript, and X.W.C., W.P.G., C.J., and L.Y. contributed to revisions of the manuscript. J.H. contributed to the fruitful discussion of this project. This manuscript was mainly prepared by P.B. and revised by L.Y., all authors participated in the manuscript preparation and commented on the manuscript.

## Competing interests

The authors declare no competing interests.
