## [Peer Review File · Nature Communications]

Donor-acceptor bulk-heterojunction sensitizer for efficient solid-state infrared-to-visible photon up-conversionEditorial Note: Parts of this Peer Review File have been redacted as indicated to remove third-party material where no permission to publish could be obtained.

REVIEWER COMMENTS

Reviewer #1 (Remarks to the Author):

In this work, Yang and coworkers successfully fabricate TTA-UC devices inspired by donor/acceptor bulk heterojunctions in organic photovoltaics, where efficient charge separation in the sensitizer donor/acceptor moiety results in efficient TTA-UC. This strategy results in bright UC emission in these devices but can also be furthered into larger scale films and impressively, onto flexible substrates. While two notable works have used bulk heterojunctions of organic semiconductors in TTA-UC schemes, (<https://doi.org/10.1038/s41566-021-00904-w> and <https://doi.org/10.1021/acsnano.3c06955>) the novelty of this work lies in the use of a donor/acceptor scheme and the higher observed efficiency of the TTA-UC process.

While the work here is novel and reported efficiencies are high, I believe there are some flaws in data analysis and interpretation that should be dealt with before publication. My largest issue is with the measurement of PLQE in this work. On line 147 and beyond, the authors argue "Typically, an integrating sphere is usually used for measuring the PLQE of a material. However, in such type devices, direct PLQE measured using an integrating sphere can be affected by self-absorption of UC photons by other components. Therefore, according to previous reports, a relative method was used to estimate the device Φ_{UC} ." As the authors compare their obtained efficiency favorably against other NIR-to-visible solid-state devices, how their value was obtained should be compared to the others. In all cases, besides the previous work from Hiramoto and coworkers, the values reported were obtained by using an integrating sphere or using a scanning microPL method with corrections. Additionally, Ref 8 in the supporting materials does not attempt to measure a Φ_{UC} , and uses a rough relative measurement to directly compare two films. Therefore, this broad comparison is faulty, in my opinion, and the films measured in this work should only be compared to those measured by Hiramoto and coworkers if any comparison is to be made at all. The other bulk heterojunction sensitized work noted in the previous paragraph, by Congreve and coworkers (Ref. 9 in the SI) measure much lower PLQE values, but they also note that they measure the neat rubrene PLQE of ~5%, which is in direct opposition to line 152 in this manuscript. An order of magnitude difference is very large in this context. While self-absorption is an issue in measuring these devices, and likely leads to lower measured values as a result of outcoupling losses (I refer the authors to a nice discussion of outcoupling/measured PLQEs here: <https://doi.org/10.1021/acsenerylett.0c01150>), the solid-state films will scatter light differently, resulting in differences in light detection. The authors should attempt to measure the absolute PLQE therefore, and can compare the two values for an estimation in outcoupling if needed, or should at least should attempt to change the refractive index of the surrounding medium to see the impact (see <https://doi.org/10.1039/C6MH00413J> for example). But as the manuscript stands, the comparisons they make to other TTA-UC devices are not valid in my opinion.

On line 176 and beyond, the authors claim that their low observed I_{th} values are a result of high charge density generation by the sensitizers. This result isn't immediately clear to me. The I_{th} can be represented formulaically, as shown originally by Monguzzi and coworkers (<https://doi.org/10.1103/PhysRevB.78.195112>). In this equation, shown below, the I_{th} depends on the rate of triplet decay in the annihilator, the efficiency of energy transfer from sensitizer to annihilator, the absorption coefficient of the sensitizer, and the second order

rate constant of the TTA process.

$$I_{th} = \left[\frac{k_{A^*T}}{\Phi_{ET} \alpha(E) \gamma_{TT}} \right]^2$$

The argument of the authors, therefore, could be argued in the context of energy (charge) transfer, or, more simply, could be just due to more light being absorbed by the sensitizer. Either way, if the authors want to contextualize their low I_{th} values, a more thorough discussion through the lens of these seminal works may be more useful.

From line 114 to 127 and 286 to 303, the authors make OPV-type devices with rubrene. Especially in the earlier section, it is unclear why this is done. The addition of rubrene makes the devices worse OPV devices, and the authors state that they have made similar OPV devices (sans rubrene), so it is unclear what this adds to the work.

In multiple places, the authors describe differences in relative intensity between 0-0 and 0-1 peaks (see line 145 for example). They firstly attribute this to absorption of UC emission by PBQx-TCl. This could also be explained, however, by differences in rubrene crystallization, as previously shown by Nienhaus and coworkers (<https://doi.org/10.1021/acs.jpcc.0c05290>). The authors thoroughly characterize the crystallization of these films compared to neat films later in the manuscript, but neglect the fact that rubrene and the sensitizer molecules could affect the crystallization of the other, resulting in differences in the rubrene crystallization, affecting the absorption/emission of the blend films as compared to the neat films.

The authors argue that the charge transfer state results in a late time feature near 820 nm in the transient absorption data shown in Figure 4. They further go on to use OPV cells of the donor/acceptor moieties and the UC device to show that the charge transfer state is the same energy, in the spectral region shown in the TA spectral map. However, no CT state is shown in Figure S10c. Why is this?

While not completely necessary, I believe some clarifying points and differences in framing would help this manuscript be more readable to a wider audience. Some suggestions are found below:

On line 37, the authors state that NIR-to-visible UC could be useful to “photodynamic therapy and so on.” In the place of so on, it might be useful to briefly discuss the benefits of NIR-to-vis UC for optogenetics.

On line 39, the authors discuss the benefits of TTA-UC, but could also add in that TTA-UC requires much lower fluences than other UC mechanisms (UCNPs for example), making devices much more feasible. The need for solid-state devices to couple with photovoltaics for example could also be stated more clearly.

The naming conventions of the components of the TTA-UC system could be improved for clarity. Typically, sensitizer and annihilator are used in TTA-UC systems, but some authors use donors/acceptors in the place of sensitizer and annihilator. This is further confused by the device described here, as a donor-acceptor BHJ is used as the sensitizer. Additionally, the authors usually refer to the annihilator as the emitter or emitting layer, but use DBP, which some authors refer to as an emitter in concert with rubrene annihilators. To avoid confusion, I'd call PYIT1:PBQx-TCl the sensitizer, rubrene the annihilator and DBP the emitter.

The authors note that DBP is often used to improve OLED and UC devices, but this might be clarified for a general audience to be less ambiguous. I'd refer the authors to excellent work by Bossanyi et al. to understand the underlying mechanism of this PLQE increase in TTA-UC systems. (<https://doi.org/10.1039/D1TC02955J>)

The authors don't discuss the impact of spin-statistics on this process until line 275. This should be introduced earlier in the manuscript, as the outline of the process on line 78-79 makes it seem like spin statistical impacts are none.

Reviewer #2 (Remarks to the Author):

Bi et al. report a solution-processed bulk heterojunction sensitizer for efficient solid-state NIR to visible photon upconversion. The UC mechanism is based on triplet generation through charge separation and recombination at the donor/acceptor interface, proposed by Izawa et al in Nat. Photon. 2021. The novel point is a mixing donor polymer to enhance charge separation efficiency. This has a big impact on improving the UC efficiency and the film can be prepared by a single-step solution process, which leads to the large area coating of a flexible substrate. Furthermore, the mixing morphology in the film and the UC mechanism is carefully investigated by many techniques such as AFM, TEM, GIWAX, and transient absorption. I think the manuscript can be published after addressing several questions.

1. I wonder how the heavy atom effect of Br in PYIT1 affects UC efficiency. How about the UC efficiency using an acceptor without Br? If UC in a mixed solution of PYIT1 and rubrene efficiently happens, triplet forms not by charge separation and recombination at the interface but by intersystem crossing in the acceptor.

2. The UC efficiency is calculated by eq.5. Eq.5 includes absorbance. The absorbance value of the film should be provided. Normally, the UC efficiency does not include absorbance, but EQE includes absorbance. Please clearly explain how to calculate the efficiency values.

3. The authors mention “the emitter rubrene crystal” in line 75, and “ordered aggregation in solution state” in line 199. The physical properties of crystal and amorphous rubrene are totally different such as HOMO level and emission spectrum, and I think rubrene in this mixed film is amorphous. Please clearly explain the crystallinity of rubrene.

4. Please add the information on DBP concentration in the main manuscript.

5. The authors mention that PYIT1 is a donor in lines 76 and 94. But, I think PYIT1 is an acceptor. If the author intended to say PYIT1 is an energy donor for UC, it should be explained as a sensitizer because a donor in BHJ blend normally means electron donor.

Reviewer #3 (Remarks to the Author):

In this work, a high-performance organic bulk heterojunction (BHJ) is doped with rubrene and its corresponding singlet emitter molecule DBP to make solution processed films that show photon upconversion based on triplet-triplet annihilation. The authors report a very high upconversion quantum yield (QY), especially for thin film, though these QY measurements were not performed in an integrating sphere (this is not standard procedure for thin films). The best performing films contain both DBP and rubrene, even though the solar cell power conversion efficiency (PCE) drops from 31.7% to 5.2% with the addition of rubrene. While the physical and electrical characterization in this paper seem to be complete, e.g. beautiful atomic force microscopy, XRD and power conversion efficiency curves, this paper makes a strong claim that charge transfer states are involved. In particular, the authors claim that triplet excitons are created from hole transfer from the acceptor to rubrene. There is little evidence for this hypothesis. Yes, there might be charge transfer states in the 17% PCE organic solar cell, but the morphology is different once DBP and rubrene are added, as seen by the decrease in PCE. If there is hole transfer, then where is the electron transfer occurring from, in order to create the triplet exciton on rubrene? If the

triplet exciton was created on rubrene, then a clear triplet excited state absorption should be observed. This is not seen in any of the spectra, whether in the main paper or SI.

There are many papers showing that rubrene can be directly excited with the 800nm laser, e.g. by Chris Bardeen. Did the authors do control experiments showing that their excitation density for transient absorption is below this threshold for the two-photon excitation of rubrene?

REVIEWERS COMMENTS

Reviewer #1 (Remarks to the Author):

In this work, Yang and coworkers successfully fabricate TTA-UC devices inspired by donor/acceptor bulk heterojunctions in organic photovoltaics, where efficient charge separation in the sensitizer donor/acceptor moiety results in efficient TTA-UC. This strategy results in bright UC emission in these devices but can also be furthered into larger scale films and impressively, onto flexible substrates. While two notable works have used bulk heterojunctions of organic semiconductors in TTA-UC schemes, (<https://doi.org/10.1038/s41566-021-00904-w> and <https://doi.org/10.1021/acsnano.3c06955>) the novelty of this work lies in the use of a donor/acceptor scheme and the higher observed efficiency of the TTA-UC process. While the work here is novel and reported efficiencies are high, I believe there are some flaws in data analysis and interpretation that should be dealt with before publication.

Author reply: We sincerely thank your very useful suggestions and the recommended excellent literature. These have helped us in further improving the manuscript. We have revised our manuscript according to your comments and suggestions, as summarised below. Changes in the manuscript are highlighted in red.

Comment 1. My largest issue is with the measurement of photo luminescence quantum efficiency (PLQE) in this work. On line 147 and beyond, the authors argue “Typically, an integrating sphere is usually used for measuring the PLQE of a material. However, in such type devices, direct PLQE measured using an integrating sphere can be affected by self-absorption of UC photons by other components. Therefore, according to previous reports, a relative method was used to estimate the device Φ_{UC} .” As the authors compare their obtained efficiency favorably against other NIR-to-visible solid-state devices, how their value was obtained should be compared to the others. In all cases, besides the previous work from Hiramoto and coworkers, the values reported were obtained by using an integrating sphere or using a scanning microPL method with corrections. Additionally, Ref 8 in the supporting materials does not attempt to measure a Φ_{UC} , and uses a rough relative measurement to directly compare two films. Therefore, this broad comparison is faulty, in my

opinion, and the films measured in this work should only be compared to those measured by Hiramoto and coworkers if any comparison is to be made at all.

Author reply: We appreciate and agree with your comments. The absolute and relative methods are two techniques used to determine PLQE. For the most of TTA-UC systems, due to reabsorption and other outcoupling losses, the UC efficiency (Φ_{UC}) is typically measured using relative method. [*Chem. Rev.* **2015**, *115*, 395] However, the Φ_{UC} values obtained from different relative methods may vary. Moreover, results obtained within different laboratories using the same relative method may also vary due to the sensitivity of measurements to the optical setups, making direct comparisons less rigorous. In this work, our main focus is not solely on achieving relatively high TTA-UC device performance, but rather on highlighting the advantages of our proposed donor-acceptor (D-A) BHJ sensitizer over a single component sensitizer in terms of charge generation and transport. Under strict identical measurement conditions, we compared the performance of different TTA-UC devices in our work. By using the D-A BHJ sensitizer, we achieved a significant enhancement in device performance compared to the single-component sensitizer. These results effectively demonstrate the effectiveness of the D-A BHJ sensitizer strategy. To ensure rigor of the paper, we have removed the comparison between different reports (**Table S3**) in the revised manuscript. We also added some explanations in the revised manuscript. The detailed modifications are shown in the **Page 6** of the revised manuscript and as follows:

“It should be noted that the Φ_{UC} results obtained using relative methods can be highly sensitive to the condition of the optical setup, leading to significant variation in the reported values.⁴⁹ Therefore, it is difficult to directly compare results from different reports. However, we believe that the TTA-UC systems reported here should be considered among the highest-performing TTA-UC systems currently available.”

Comment 2. The other bulk heterojunction sensitized work noted in the previous paragraph, by Congreve and coworkers (Ref. 9 in the SI) measure much lower PLQE values, but they also note that they measure the neat rubrene PLQE of ~5%, which is in direct opposition to line 152 in this manuscript. An order of magnitude difference is very large in this context.

Author reply: We thank the reviewer for highlighting this point. In principle, the PLQE measurement values of a material should be relatively consistent. However, many factors can influence the PLQE results, such as the measurement method, the purity of the material, the sample preparation method, and even the degradation effect of the samples during measurement. [Adv. Mater. **1997**, 9, 230] Therefore, different PLQE measurement results often exhibit deviations. For example, the PLQE measurement results for PFO range from 45% to 80% [Chem. Phys. Lett. **2011**, 506, 321]. Similarly, the PLQE measurement values for DBP in solution state range from 40% to 80% [J. Phys. Chem. A **2023**, 127, 99]. According to the previous reports, we also found diverse measurement results for rubrene film. For instance, Congreve et al. measured a PLQE of less than 5% for rubrene film which was prepared by spin-coating method [ACS Nano **2023**, 17, 22642]. Kazlauskas et al. obtained a rubrene PLQE of 20.5% by preparing rubrene films using the “cold deposition mode” and “post-annealing” methods [J. Mater. Chem. C, **2022**, 10, 6314]. Additionally, Mori et al. achieved a PLQE of 25% for neat rubrene porous film [ACS Macro Lett. **2023**, 12, 523]. Furthermore, Hiramoto et al. measured the PLQE of rubrene films prepared by thermal evaporation to be around 40% [Nat. Photon. **2021**, 15, 895]. However, due to the energetics of the singlet and triplet states of rubrene ($E_S \approx 2E_T$), singlet fission in solid-state films often significantly reduces rubrene’s singlet emission, leading to low PLQE values. Some high PLQE results for rubrene may be attributed to some factors that partially inhibit singlet fission or are due to variations in measurement and calculation methods.

According to literature, the reported variation in PLQE results for rubrene:DBP blend films with appropriate DBP doping ratios are relatively small, typically ranging from 40% to 50%, except for the measurement of 71.2% by Hiramoto et al. [Nat. Photon. **2021**, 15, 895] Here, to demonstrate the comparability of our results, we replaced the reference sample (rubrene) with a rubrene:DBP (1:0.75%, weight ratio) blend film. Employing the de Mello method, we obtained an average PLQE value of $48.98 \pm 2.02\%$ for rubrene:DBP film, which is similar with some other reported values. We then used the same relative method as before, and recalculated the Φ_{UC} values for PYIT:PBQx-TCl:rubrene-, PYIT:PBQx-TCl:rubrene:DBP- and PYIT:rubrene-based devices, which are 0.85%, 2.20% and 0.09%, respectively. The Φ_{UC} of PYIT1:rubrene UC device shown here is comparable to the values reported by Congreve ($EQE \approx 0.02\%$) [ACS Nano **2023**, 17, 22642], with some discrepancies likely due to the matching of our sensitizer absorption spectra

and excitation wavelength, as well as differences in material properties. Although the values slightly decreased compared to the original, they enhance the comparability of our results with those of other studies. The results have also demonstrated the effectiveness of our D-A BHI sensitizer in preparing high-performance TTA-UC systems. We appreciate your suggestion again. We have added the modified data to the revised manuscript, and the detailed modifications are shown in **Page 6** of the revised manuscript and as follows:

“The saturated Φ_{UC} for the PYIT1:PBQx-TCl:rubrene device is 0.85%. We also measured the performance of the PYIT1:rubrene TTA-UC device. As shown in Fig. S6, the Φ_{UC} of the PYIT1:rubrene device is 0.09%.”

Comment 3. While self-absorption is an issue in measuring these devices, and likely leads to lower measured values as a result of outcoupling losses (I refer the authors to a nice discussion of outcoupling/measured PLQEs here: <https://doi.org/10.1021/acsenergylett.0c01150>), the solid-state films will scatter light differently, resulting in differences in light detection. The authors should attempt to measure the absolute PLQE therefore, and can compare the two values for an estimation in outcoupling if needed, or should at least should attempt to change the refractive index of the surrounding medium to see the impact (see <https://doi.org/10.1039/C6MH00413J> for example). But as the manuscript stands, the comparisons they make to other TTA-UC devices are not valid in my opinion.

Author reply: We thank the reviewer’s useful suggestions. We agree with the reviewer that the absolute method is indeed widely proven to give an underestimate of Φ_{UC} due to the outcoupling losses. In fact, we have attempted to measure the absolute PLQE of UC emission using the de Mello method before. However, at lower excitation intensities, although bright UC emission could be visibly observed in the integrating sphere, we could hardly collect an effective UC emission spectrum using the spectrometer due to issues such as reabsorption and scattering etc. Further increasing the excitation intensity led to saturation of our spectrometer, making it difficult to accurately calculate the number of incident photons. We attribute the major difficulty to the severe reabsorption of UC emission by sensitizer molecules in the integrating sphere.

According to literature, we roughly estimated the reabsorption effect by comparing the UC emission spectra of the PYIT1:PBQx-TCl:rubrene:DBP-based TTA-UC system with and without the integrating sphere. [*J. Phys. Chem. A* **2019**, *123*, 10197] The schematic diagram is shown in **Fig. R1a**. In Experiment A, the sample was placed inside the integrating sphere, and the laser beam was directed onto the UC film. Experiment B was similar to Experiment A except that the integrating sphere was removed without changing other configurations. The UC emission spectrum obtained in Experiment A represents the emission after reabsorption, while the emission spectrum obtained by Experiment B is the UC spectrum with very weak reabsorption (**Fig. R1b**). Comparing the photons obtained from emission spectra A and B, the effect of reabsorption is particularly severe for our TTA-UC device, resulting in nearly an order of magnitude outcoupling loss. We briefly analysed the reasons for the large outcoupling loss. Firstly, the broad absorption of PYIT1 and PBQx-TCl in our D-A BHJ sensitizer significantly overlaps with the UC emission, as shown in **Fig. R1c**. Additionally, both PYIT1 and PBQx-TCl have large absorption coefficients of 1.9×10^5 and $1.0 \times 10^5 \text{ M}^{-1} \text{ cm}^{-1}$, respectively (**Fig. R1d-g**). Furthermore, the ratio of sensitizer to annihilator is relatively high (weight ratio of 1:2.5), and the thickness of the UC film is over 100 nm, which results in strong reabsorption of sensitizer for UC emission. Secondly, factors such as light scattering also contribute to other coupling losses. We have added the new **Fig S4 (Fig R1 here)**, along with explanations in the revised manuscript. The detailed modifications are shown in **Page 5** of the revised manuscript and as follows:

Fig. R1. **a-b** Experimental configurations for collecting UC spectra A and B. **c** The normalised absorption and UC emission spectra of the PYIT1:PBQx-TCI:rubrene:DBP film. **d-g** Absorption spectra of PYIT1 and PBQx-TCI with different concentrations in solutions and the fitted absorption coefficients.

“Typically, an integrating sphere is usually used for measuring the absolute photoluminescence quantum efficiency (PLQE) of a material. However, for TTA-UC devices, direct PLQE measured using an integrating sphere can be affected by reabsorption of UC photons by sensitizers.⁷ We estimated the reabsorption effect by comparing the UC emission spectra of the PYIT1:PBQx-TCI:rubrene:DBP-based TTA-UC system with (A) and without (B) the integrating sphere (Fig. S4).⁴⁷ The UC emission spectrum obtained in Experiment A represents the spectrum after reabsorption, while the emission spectrum outside the sphere in Experiment B can be considered as the UC emission with minimum reabsorption effect. Comparing the photons obtained from emission spectra A and B, the effect of reabsorption is particularly severe for our TTA-UC device, resulting in nearly an order of magnitude outcoupling loss. This is due to the significant overlap between our UC emission and the absorption spectra of the D-A sensitizer. Additionally, as shown

in Fig. S4d-g, both PYIT1 and PBQx-TCI exhibit relatively large absorption coefficients, at 1.9×10^5 and $1.0 \times 10^5 \text{ M}^{-1} \text{ cm}^{-1}$, respectively. Besides, experimental output losses such as waveguiding, scattering, and inner-filter effects can also induce outcoupling losses. Therefore, a relative method was used to estimate the device Φ_{UC} .”

Comment 4. On line 176 and beyond, the authors claim that their low observed I_{th} values are a result of high charge density generation by the sensitizers. This result isn’t immediately clear to me. The I_{th} can be represented formulaically, as shown originally by Monguzzi and coworkers (<https://doi.org/10.1103/PhysRevB.78.195112>). In this equation, shown below, the I_{th} depends on the rate of triplet decay in the annihilator, the efficiency of energy transfer from sensitizer to annihilator, the absorption coefficient of the sensitizer, and the second order rate constant of the TTA process.

$$I_{th} = \frac{(k_A^T)^2}{\phi_{tr} \alpha(E) \gamma_{TT}}$$

The argument of the authors, therefore, could be argued in the context of energy (charge) transfer, or, more simply, could be just due to more light being absorbed by the sensitizer. Either way, if the authors want to contextualize their low I_{th} values, a more thorough discussion through the lens of these seminal works may be more useful.

Author reply: Thanks for the reviewer’s very useful suggestions. In this work, we obtained relatively low I_{th} values and we simply attributed to the higher carrier density. This explanation may not be comprehensive enough to allow readers to directly understand. I_{th} is an important parameter characterizing the TTA-UC processes, providing insight into the requisite incident power densities for maximizing Φ_{UC} . Generally, I_{th} is directly affected by the rate of spontaneous decay of the acceptor, the absorption coefficient of the sensitizer, the efficiency of triplet energy transfer (TET), and the rate of TTA. This relationship can be expressed in **Eq. R1**.

$$I_{th} = \frac{(k_A^T)^2}{\alpha(E) \gamma_{TT}} \left(\frac{k_{tr}}{k_{tr} + k_D^T} \right) \equiv \frac{(k_A^T)^2}{\phi_{tr} \alpha(E) \gamma_{TT}} \quad \mathbf{R1,}$$

where α is the absorption coefficient at the excitation wavelength, γ_{TT} is the second-order annihilation constant for the TTA. k_A^T is the rate of triplet decay in the annihilator. Therefore, to obtain a low I_{th} , it is crucial that the sensitizer has a strong absorption capability at the excitation

wavelength. As shown in **Fig. R1e**, the PYIT1 exhibits a strong molar extinction coefficient at 808 nm ($1.9 \times 10^5 \text{ M}^{-1} \text{ cm}^{-1}$). This ensures that the TTA-UC system possesses sufficient photon absorption. Secondly, the sensitizer and annihilator should have a high TET efficiency. Compared to traditional TTA-UC systems, our TTA-UC device does not rely on intersystem crossing (ISC) or energy transfer processes but instead directly utilizes charge transfer and charge recombination processes. Morphologically, rubrene is well-dispersed within the fiber network of PYIT1 and PBQx-TCl, ensuring efficient charge transfer and significantly reducing non-radiative triplet recombination during the energy transfer process in traditional TTA-UC systems. Therefore, under low-power excitation, a large charge carrier density can be generated through charge recombination, leading to efficient triplet accumulation in the rubrene domains. Furthermore, the annihilator triplet should have a long lifetime. We also measured the UC emission lifetime using time-resolved photoluminescence (TRPL) with an excitation wavelength of 808 nm. The UC emission lifetime of the PYIT1:PBQx-TCl:rubrene-based film is 22 μs (**Fig. R2**), slightly longer than that of the same type TTA-UC devices. We appreciate the suggestions from the reviewer. In order to provide clearer explanations for the readers, we have added the new **Fig S8 (Fig R2 here)** and incorporated additional clarifications in the revised manuscript. The detailed modifications are presented on **Page 6** of the revised manuscript and are as follows:

“For traditional TTA-UC systems, I_{th} can be expressed by the equation $I_{th} = (\alpha \Phi_{TET} k_{TTA} \tau_{TE}^2)^{-1}$.

⁵¹ The PYIT1 possesses a high molar extinction coefficient of $1.9 \times 10^5 \text{ M}^{-1} \text{ cm}^{-1}$. Furthermore, compared to the traditional TTA-UC systems that rely on ISC and energy transfer processes, the working mechanism of our device is based on efficient charge generation and recombination processes. This significantly reduces the non-radiative energy losses associated with triplet diffusion and energy transfer processes in traditional TTA-UC devices. This ensures the accumulation of a high density of triplet excitons within the rubrene domains even under low-energy excitation. Additionally, we also measured the lifetime of the UC emission, which was found to be 22 μs (Fig. S8). These highly efficient processes enable the achievement of low threshold power densities for our PYIT1:PBQx-TCl:rubrene and PYIT1:PBQx-TCl:rubrene:DBP UC devices, with I_{th} values of 8 and 10 mW cm^{-2} , respectively. These I_{th} values are among the lowest reported to date, indicating a very low threshold power needed to elicit efficient UC.”

Fig. R2. TRPL decay curve of PYIT1:PBQx-TCl:rubrene film with excitation of 800 nm.

Comment 5. From line 114 to 127 and 286 to 303, the authors make OPV-type devices with rubrene. Especially in the earlier section, it is unclear why this is done. The addition of rubrene makes the devices worse OPV devices, and the authors state that they have made similar OPV devices (sans rubrene), so it is unclear what this adds to the work.

Author reply: We appreciate the reviewer's comments. To clarify, the point here is not to achieve better OPV devices with rubrene added. In this work, we proposed and utilised a D-A BHJ sensitizer to fabricate efficient TTA-UC devices. Therefore, a high-performance BHJ is essential as it facilitates efficient charge transfer and transport, effectively suppressing both geminate and non-geminate recombination processes. To evaluate the PYIT1:PBQx-TCl-based BHJ performance, we initially fabricated OPV devices, as they are devices that can convert photons into charge carriers, providing a good measure of charge generation and recombination. Additionally, considering that our TTA-UC devices operate under monochromatic light, we also evaluated the device performance under laser illuminations. The results showed that our PYIT1:PBQx-TCl-based OPV devices exhibited good performance under both AM 1.5G (100 mW cm^{-2}) and 808 nm laser [*Nat. Commun.* **2023**, *14*, 5511], indicating that PYIT1:PBQx-TCl would be a suitable BHJ option in terms of performance. Furthermore, we prepared the PYIT1:PBQx-TCl:rubrene-based OPV devices to demonstrate that efficient charge transfer still occurs between PYIT1:PBQx-TCl even with the addition of a large amount of rubrene. Although there was an obvious decrease in PCEs under both AM 1.5G and 808 nm laser, the open-circuit voltage (V_{oc}) only decreased from 0.92 V to 0.90 V. We also fabricated PYIT1:rubrene-based

devices and measured the performance under AM 1.5G (100 mW cm⁻²) illumination. As shown in **Fig. R3**, the device shows a PCE of 0.10% with a V_{OC} of 0.78 V, J_{SC} of 0.38 mA cm⁻² and FF of 35%. According to previous reports, weak charge transfer between donor and acceptor leads to severe geminate recombination, resulting in significant V_{OC} losses. [*Nat. Mater.* **2018**, *17*, 119; *Joule* **2018**, *2*, 25] Therefore, based on the V_{OC} loss observed in PYIT1:rubrene-based devices (0.64 eV), it can be inferred that efficient charge transfer does not occur between PYIT1 and rubrene. Consequently, this comparison suggests that in the PYIT1:PBQx-TCl:rubrene system (where V_{OC} almost did not change), the majority of charge carriers generated by PYIT1 are still transferred to PBQx-TCl. Transient absorption (TA) spectroscopy results also confirm this conclusion. In both the PYIT1:PBQx-TCl and PYIT1:PBQx-TCl:rubrene films, when excited at 800 nm with only PYIT1 being excited, we can clearly observe the hole transfer signals from PYIT1 to PBQx-TCl (**Fig. R4**). The results indicate that the BHJ structure based on PYIT1:PBQx-TCl as a sensitizer efficiently provides free charge carriers. To make the purpose of preparing OPV devices more directly understandable to readers, we have added the new **Fig S3 (Fig R3 here)**, and some explanations in the revised manuscript. The detailed modifications are shown in **Page 4** of the revised manuscript and as follows:

“For the D-A BHJ sensitizer, a good BHJ is crucial as it enables efficient charge transfer and transport, effectively suppressing geminate and non-geminate recombination. To evaluate the BHJ performance, we initially fabricated organic photovoltaic (OPV) devices, as they are providing a good measure of charge generation and recombination.”

“Additionally, considering that the TTA-UC devices often operate under monochromatic light, we also evaluated the device performance using an 808-nm laser as the light source.”

“Comparing the large V_{OC} loss observed in PYIT1:rubrene-based OPV cell (0.64 V) (Fig. S3), we reasonably speculate that charge separation primarily occurs between PYIT1 and PBQx-TCl in PYIT1:PBQx-TCl:rubrene BHJ.”

Fig. R3. The current-density-voltage (J - V) curve and device parameters of the PYIT1:rubrene-based OPV cell under AM 1.5G (100 mW cm^{-2}).

Fig. R4. The TA images of (a) PYIT1:PBQx-TCl and (b) PYIT1:PBQx-TCl:rubrene films under 800 nm excitation.

Comment 6. In multiple places, the authors describe differences in relative intensity between 0-0 and 0-1 peaks (see line 145 for example). They firstly attribute this to absorption of UC emission by PBQx-TCl. This could also be explained, however, by differences in rubrene crystallization, as previously shown by Nienhaus and coworkers (<https://doi.org/10.1021/acs.jpcc.0c05290>). The authors thoroughly characterize the crystallization of these films compared to neat films later in the manuscript, but neglect the fact that rubrene and the sensitizer molecules could affect the crystallization of the other, resulting in differences in the rubrene crystallization, affecting the absorption/emission of the blend films as compared to the neat films.

Author reply: Thanks for the reviewer's suggestions. In the manuscript, we discussed the relative intensity changes of different peaks in absorption spectra and PL spectra. Firstly, in **Fig. S1**, we

discussed the changes in the absorption peak intensities of the 0-0 and 0-1 transitions of PBQx-TCl in the blend film after the introduction of rubrene into the PYIT1:PBQx-TCl system. This indicates a reduction in the crystalline size of PBQx-TCl and PYIT1, which is further confirmed by the GIWAXS results (**Fig 3g-i, Fig S12**). Furthermore, we observed a relative intensity change in the PL 0-0 and 0-1 peaks of rubrene in the UC film compared to rubrene in the neat film (**Fig. 1c**). In the manuscript, we briefly attributed this to potential absorption issues with PBQx-TCl; however, we agree with the reviewer this explanation may indeed not be comprehensive enough. As the reviewer pointed out, differences in molecular crystallinity can significantly affect the optical properties of materials. In some literature [*Adv. Mater.* **2011**, *23*, 5370; *Phys. Rev. B* **2013**, *87*, 201203; *Phys. Rev. B* **2014**, *90*, 205305], we also observed that the PL spectra of rubrene exhibit peak shifts and changes in peak intensities depending on the rubrene crystallinity, as illustrated in **Fig. R5**. In our GIWAXS results, we observed obvious changes in the diffraction pattern of rubrene in the blend film compared to the neat film. Many diffraction spots disappeared, indicating a notable change in the crystallinity of rubrene. Combining the relevant literature mentioned above, the differences in relative intensity between 0-0 and 0-1 emission peaks of rubrene should be attributed to changes in its crystallinity. We have added related explanations in the revised manuscript. The detailed modifications are shown on **Page 5** of the revised manuscript and as follows:

[Redacted]

Fig. R5. Some examples of PL spectra of rubrene in different crystalline states from the references of (a) *Adv. Mater.* **2011**, *23*, 5370, (b) *Phys. Rev. B* **2013**, *87*, 201203, and (c) *Phys. Rev. B* **2014**, *90*, 205305.

“Compared to the PL spectra of the neat rubrene film, we observed a variation in the relative intensity between the 0-0 and 0-1 peaks in the UC emission spectra. According to previous reports, this can be attributed to the change in the crystalline state of rubrene from the neat film to the blend film.^{45, 46}”

Comment 7. The authors argue that the charge transfer state results in a late time feature near 820 nm in the transient absorption data shown in Figure 4. They further go on to use OPV cells of the donor/acceptor moieties and the UC device to show that the charge transfer state is the same energy, in the spectral region shown in the TA spectral map. However, no CT state is shown in Figure S10c. Why is this?

Author reply: Thank you for the reviewer’s comments. To clarify, the CT state energy comparison using OPVs is referring to the CT state generated between PYIT1 and PBQx-TCl; whereas the 820 nm signal is associated with the subsequent intermediate state between PYIT1 and rubrene. We observed photoinduced absorption (PIA) signals near 820 nm in the TA spectra of both PYIT1:rubrene and PYIT1:PBQx-TCl:rubrene systems, whereas this signal was not observed in the PYIT1:PBQx-TCl system (**Fig. S14**). Combining other optoelectronic and optical measurement results, we reasonably attribute this signal to intermediate-state absorption related to charge recombination between PYIT1 and rubrene, as described in the manuscript (**Page 10**).

In the manuscript, we demonstrate that the TTA-UC process initiates with the exciton separation at PYIT1:PBQx-TCl interface. OPV cells based on PYIT1:rubrene and PYIT1:PBQx-TCl:rubrene were fabricated to assess the charge generation, recombination and transport properties. The CT states play a crucial role in exciton dissociation and charge recombination processes. Through fitting the high-sensitivity EQE (HEQE) and EL spectra of the OPV devices, we obtained similar energies for the CT state in both the PYIT1:PBQx-TCl and PYIT1:PBQx-TCl:rubrene systems. This indicates that in both systems, the excitons generated by PYIT1 are primarily separated at the interface of PYIT1 and PBQx-TCl. As the reviewer mentioned, we should observe absorption signals corresponding to CT state in the spectra. In previous studies, TA spectroscopy enabled the clear observation of PIA signals corresponding to localized excitons (LE), CT excitons, and charge-separated (CS) states [*J. Am. Chem. Soc.* **2020**, *142*, 12751]. Here, we further measured the TA spectra in the NIR region with 800 nm excitation, and the results are

shown in **Fig. R6**. According to the literature, in the spectrum of PYIT1:PBQx-TCl, we observed distinct absorption of the PYIT1 LE at 850 nm, and at 940 nm, absorption corresponding to the CT exciton between PYIT1 and PBQx-TCl was also observed [*J. Am. Chem. Soc.* **2021**, *143*, 20, 7599]. The kinetics curves also exhibit characteristics consistent with those of LE and CT excitons. In the TA spectra of PYIT1:PBQx-TCl:rubrene, we observed absorption signals corresponding to the LE of PYIT1 and the CT exciton between PYIT1 and PBQx-TCl at the same positions. These results suggest that under 800 nm excitation, after the generation of excitons by PYIT1, holes initially transfer to PBQx-TCl.

Fig. R6. The TA images of (a) PYIT1:PBQx-TCl and (b) PYIT1:PBQx-TCl:rubrene films. TA spectra of (c) PYIT1:PBQx-TCl and (d) PYIT1:PBQx-TCl:rubrene films at different delay times. The normalised decay curves probed at various wavelengths recorded from (e) PYIT1:PBQx-TCl and (f) PYIT1:PBQx-TCl:rubrene films.

Fig. R7. The normalised decay curves probed at 580 nm and 615 nm recorded from (a) PYIT1:PBQx-TCl and (b) PYIT1:PBQx-TCl:rubrene films.

Additionally, in the PYIT1:PBQx-TCl-based TA spectrum in visible region, we also extracted absorption signal at 615 nm. According to literature, this signal corresponds to charge-separated (CS) states between PYIT1 and PBQx-TCl. We extracted their dynamic processes, as shown in **Fig. R7**. This decay dynamics align with the reported dynamic processes in the literature. [*J. Am. Chem. Soc.* **2021**, *143*, 20, 7599.] Importantly, we also observed the same signal in the TA spectra of the PYIT1:PBQx-TCl:rubrene system, indicating the presence of these CT and CS states (**Fig R7**). This further confirms our description that photogenerated excitons separate at the PYIT1:PBQx-TCl interface, followed by subsequent TTA-UC processes. We have included additional figures (**Fig R6** as the new **Fig S16**, and **Fig R7** as the new **Fig S17**) and explanations in the revised manuscript. The detailed modifications are shown in **Page 10** of the revised manuscript and as follows:

“The TA spectra in the NIR region with 800 nm excitation were also measured to demonstrate the generation of CT states between PYIT1 and PBQx-TCl, and the results are shown in Fig. S16. For PYIT1:PBQx-TCl, we observed distinct absorption of the PYIT1 localized exciton (LE) at 850 nm, and at 940 nm, absorption corresponding to the CT exciton between PYIT1 and PBQx-TCl is also observed.⁶¹ The decay kinetics of 850 nm and 940 nm signals also exhibit characteristics consistent with those of LE and CT excitons. For the PYIT1:PBQx-TCl:rubrene system, we observed absorption signals corresponding to the LE of PYIT1 and the CT exciton between PYIT1 and PBQx-TCl at the similar positions. Additionally, in both of the PYIT1:PBQx-TCl and PYIT1:PBQx-TCl:rubrene TA spectra in visible region, we also extracted absorption signals at

615 nm. According to previous report, these signals correspond to charge-separated (CS) states between PYIT1 and PBQx-TCI.⁶¹ The decay dynamics of the CS states are shown in Fig. S17. The results further confirm our description that photogenerated excitons initially separated at the PYIT1:PBQx-TCI interface, followed by subsequent TTA-UC processes.”

Comment 8. While not completely necessary, I believe some clarifying points and differences in framing would help this manuscript be more readable to a wider audience. Some suggestions are found below:

Author reply: We greatly appreciate the reviewer’s attention to these detailed issues. Following your suggestions, we have made careful revisions.

- a. On line 37, the authors state that NIR-to-visible UC could be useful to “photodynamic therapy and so on.” In the place of so on, it might be useful to briefly discuss the benefits of NIR-to-vis UC for optogenetics.

Author reply: Optogenetics is a technology enabling light-sensitive proteins to regulate cellular processes. Utilising the superior tissue penetration of NIR light compared to visible light, NIR-to-Vis TTA-UC presents significant advantages as the light source for optogenetics and some other various biological applications. We have included the corresponding discussions in the revised manuscript. The detailed modifications are shown in **Page 1** of the revised manuscript and as follows:

“Additionally, utilising the superior tissue penetration of NIR light compared to visible light, NIR-to-visible presents significant advantages as an internal light source for various biological applications such bioimaging, drug delivery, photodynamic therapy, and optogenetics.^{10-13”}

- b. On line 39, the authors discuss the benefits of TTA-UC, but could also add in that TTA-UC requires much lower fluences than other UC mechanisms (UCNPs for example), making devices much more feasible. The need for solid-state devices to couple with photovoltaics for example could also be stated more clearly.

Author reply: We have made changes accordingly, detailed revisions can be found on **Page 1** of the revised document, outlined as follows:

“Solid-state UC device is conducive to robust integration into in a wide variety of solar device architectures.”

“Besides, TTA-UC system possesses low excitation power requirements and easily tunable excitation and emission wavelengths.”

- c. The naming conventions of the components of the TTA-UC system could be improved for clarity. Typically, sensitizer and annihilator are used in TTA-UC systems, but some authors use donors/acceptors in the place of sensitizer and annihilator. This is further confused by the device described here, as a donor-acceptor BHJ is used as the sensitizer. Additionally, the authors usually refer to the annihilator as the emitter or emitting layer, but use DBP, which some authors refer to as an emitter in concert with rubrene annihilators. To avoid confusion, I'd call PYIT1:PBQx-TCI the sensitizer, rubrene the annihilator and DBP the emitter.

Author reply: Thank you for your suggestion. To prevent reader confusion, we have made corresponding changes in the revised manuscript, referring to PYIT1:PBQx-TCI as the sensitizer, rubrene as the annihilator, and DBP as the emitter.

- d. The authors note that DBP is often used to improve OLED and UC devices, but this might be clarified for a general audience to be less ambiguous. I'd refer the authors to excellent work by Bossanyi et al. to understand the underlying mechanism of this PLQE increase in TTA-UC systems. (<https://doi.org/10.1039/D1TC02955J>)

Author reply: Due to the energetics of the triplet state, which is approximately half the energy of the singlet state, rubrene can engage in both TTA and singlet fission (SF). Consequently, DBP is often introduced into the rubrene layer to efficiently capture singlet states from rubrene, thereby suppressing SF and enhancing the luminous efficiency of the corresponding devices. [*J. Mater. Chem. C*, **2022**, *10*, 4684.] Following your suggestion, we have included the purpose and concentration of DBP used in the manuscript. The detailed modifications are shown in **Page 6** of the revised manuscript and are as follows:

“Due to the energetics of the triplet state, which is approximately half the energy of the singlet state, rubrene can engage in both TTA and singlet fission (SF). Consequently, tetraphenyldibenzoperiflanthene (DBP) is often introduced into the rubrene layer to efficiently capture singlet states from rubrene, thereby suppressing SF and enhancing the corresponding device performance.⁴⁸ The molecular structure, absorption and PL spectra of DBP are shown in Fig. S7. Here, we introduce DBP into the PYIT1:PBQx-TCl:rubrene system as the emitter with a weight ratio relative to rubrene of 0.75 wt% and fabricated the PYIT1:PBQx-TCl:rubrene:DBP UC device using the same one-step method.”

- e. The authors don't discuss the impact of spin-statistics on this process until line 275. This should be introduced earlier in the manuscript, as the outline of the process on line 78-79 makes it seem like spin statistical impacts are none.

Author reply: Following your suggestion, we have incorporated spin statistics into the introduction, where the charge recombination process is initially mentioned. The detailed modifications are shown in **Page 2** of the revised manuscript and are as follows:

“Subsequently, holes are transferred from PBQx-TCl to rubrene, forming charge transfer (CT) excitons between PYIT1 and rubrene following the spin statistics (75% triplets and 25% singlets).”

Reviewer #2 (Remarks to the Author):

Bi et al. report a solution-processed bulk heterojunction sensitizer for efficient solid-state NIR to visible photon up-conversion. The UC mechanism is based on triplet generation through charge separation and recombination at the donor/acceptor interface, proposed by Izawa et al in Nat. Photon. 2021. The novel point is a mixing donor polymer to enhance charge separation efficiency. This has a big impact on improving the UC efficiency and the film can be prepared by a single-step solution process, which leads to the large area coating of a flexible substrate. Furthermore, the mixing morphology in the film and the UC mechanism is carefully investigated by many techniques such as AFM, TEM, GIWAX, and transient absorption. I think the manuscript can be published after addressing several questions.

Author reply: We sincerely thank for your suggestions that led us to further improve the manuscript. We have revised our manuscript according to your comments, as summarised below. Changes in the manuscript are highlighted in red.

Comment 1. I wonder how the heavy atom effect of Br in PYIT1 affects UC efficiency. How about the UC efficiency using an acceptor without Br?

Author reply: Thanks for the reviewer's comments. Halogen atoms, such as F, Cl, Br, and I are often used to regulate the optoelectronic properties and molecular packing of molecules. [*Acc. Mater. Res.* **2021**, 2, 986] In fact, we chose PYIT1 as the sensitizer in this work not because of the Br atoms on its end group, but mainly because of the following two reasons: Firstly, our previous work has demonstrated that the PYIT1:PBQx-TCl-based cell has excellent performance with a PCE of 17.4% and a relative high fill factor of 75.9%. This indicates that the PYIT1:PBQx-TCl-based BHJ has good charge generation and transport properties. Secondly, PYIT1 has a very suitable absorption spectrum, with a peak at 806 nm, which matches the wavelength of the commonly used 808 nm continuous wave (CW) laser.

We are grateful to the reviewer for pointing out this point. Here, we conducted a separate brief yet holistic study on whether the Br atoms has a significant impact on the TTA-UC device performance. Due to the difficulty in synthesizing a Br-free PYIT1, we chose two structurally similar small molecule acceptors Y5 and Y5-2Br, differing only in the presence or absence of the

Br atoms in end groups, as shown in **Fig. R8a**. They possess similar molecular energy levels (**Fig. R8b**). The film absorption spectrum of Y5-2Br exhibits a 10 nm redshift relative to Y5 (**Fig. R8c**). Besides, as shown in **Fig. R8f**, they have similar extinction coefficients in solution states at their absorption peak, which are 1.93×10^5 and $1.97 \times 10^5 \text{ M}^{-1} \text{ cm}^{-1}$ for Y5 (720 nm) and Y5-2Br (735 nm), respectively. We also investigated their crystallinities. In both neat and blend films, Y5-2Br exhibits relatively strong molecular ordering compared to Y5 (**Fig. R9**). Additionally, we performed TA measurements on the charge transfer dynamics of the two ternary blend films. The results are similar to those obtained in this work. We can see obvious GSB signals of the PBQx-TCl (**Fig. R10a-b**). Furthermore, their hole transfer processes show no significant differences (**Fig. R10c**). Under the same preparation conditions, we measured the Φ_{UCS} of the two systems. Their efficiencies are 0.31% and 0.36%, respectively (**Fig. R10d**). From these results, we think that Br may indirectly affect device performance by influencing molecular properties, but it is unlikely to be the determining factor for TTA-UC device performance. In future studies, we will further investigate this aspect. Additionally, we have provided an explanation for our choice of the PYIT1 molecule in the manuscript. The detailed modifications are shown in **Page 3** of the revised manuscript and are as follows:

“PYIT1 is used here as the energy donor component in the sensitizer due to its decent optoelectronic properties, such as the suitable energy levels and absorption spectrum.”

Fig. R8. **a** Molecular structures of Y5 and Y5-2Br. **b** The energy levels of Y5 and Y5-2Br. **c** The normalised absorption spectra of neat Y5 and Y5-2Br. **d-f** Absorption spectra of Y5 and Y5-2Br with different concentrations in solution and the fitted absorption coefficients.

Fig. R9. **a** 2D GIWAXS patterns of the Y5 and Y5-2Br-based neat and blend films. **b** The 1D GIWAXS integration curves along IP and OOP directions.

Fig. R10. The TA images of (a) Y5:PBQx-TCl:rubrene and (b) Y5-2Br:PBQx-TCl:rubrene films. **c** Dynamics of the hole transfer in Y5:PBQx-TCl:rubrene and Y5-2Br:PBQx-TCl:rubrene films. **d** The Φ_{UC} of the Y5:PBQx-TCl:rubrene and Y5-2Br:PBQx-TCl:rubrene:DBP UC devices as function of power densities.

Comment 2. If UC in a mixed solution of PYIT1 and rubrene efficiently happens, triplet forms not by charge separation and recombination at the interface but by intersystem crossing in the acceptor.

Author reply: Thanks for the reviewer's comments. Given that Br or heavy atom effect is not the key behind the enhanced TTA-UC, we observe the UC emission of solid-state PYIT1:rubrene film is much weaker than that of the PYIT1:PBQx-TCl:rubrene film (**Fig. 2c**), and we have clearly explained the mechanisms of the UC in the solid state. Here, we shall briefly investigate the working mechanism in solution phase systems. The photons of the UC emission from rubrene in PYIT1:rubrene (1:1) solution under 700 nm illumination is shown in **Fig. R11a**. Two most likely working mechanisms are considered: **1.** As speculated by the reviewer, the formation of a triplet state in PYIT1, followed by energy transfer to rubrene; **2.** Another possibility is a process similar to the charge transfer and recombination mechanism observed in our solid-state TTA-UC devices. Firstly, if it is a mechanism based on energy transfer similar to the traditional TTA-UC system, then the triplet energy (T_1) of PYIT1 must be higher than that of rubrene. We calculated the energy levels of PYIT1 and rubrene using TD-DFT, and the results are shown in **Fig. R12** and **Table R1**. Under the same calculation conditions, the T_1 of PYIT1 is indeed higher than that of rubrene. Next, if the UC mechanism is based on energy transfer, the UC emission intensity (I_{UC}) should be highly sensitive to the molar ratio of the sensitizer to emitter (S/E). Here, taking PtDAP:rubrene (A) and PYIT1:rubrene (B) solution systems as examples, we compared the variation of I_{UC} with the S/E value. The change in I_{UC} with S/E is shown in **Fig. R11b-c**. The X-axis labels 1, 2, 3, 4, 5 represent 1:2, 1:10, 1:20, 1:40, 1:80, respectively. We can observe that the I_{UC} of system A exhibits a strong dependence on S/E, which is consistent with the previous report. [*Chem. Eur. J.* **2008**, *14*, 9846] When the ratio of S to E is 1:2, the I_{UC} is very low, and the UC emission is hardly to be observed. This is due to severe TTA caused by molecular aggregation of the sensitizer at higher concentrations, as proven by many reports. [*Chem. Soc. Rev.* **2020**, *49*, 6529] When the S/E ratio reaches 1:40, I_{UC} gradually saturates. In contrast, the I_{UC} of the PYIT1:rubrene system exhibits a completely different trend. Even when the relative concentration of the sensitizer is large, a strong relative I_{UC} can still be obtained. As the concentration of the sensitizer decreases, after reaching its maximum value, the I_{UC} then begins to slightly decrease. Additionally, it is worth noting that in OPV acceptor molecules, due to weak spin-orbit coupling

(SOC) effect, intersystem crossing (ISC) is weak and cannot generate a high density of triplet excitons through ISC, which may not lead to obvious UC emission based on the energy transfer mechanism. In conclusion, we believe that in the PYIT1:rubrene solution, weak UC emission should be attributed to the process of charge separation followed by recombination. We once again thank your comments, which are very helpful for the optimisation and application of such UC systems in solution states. In our future work, we will conduct more systematic studies on the working mechanism and performance of these TTA-UC systems in solution state.

Fig. R11. a Photograph of UC emission of PYIT1:rubrene solution under 700 nm excitation. Dependence of the I_{UC} at different S/E ratios for (b) PtDAP:rubrene and (c) PYIT1:rubrene solutions, with x-axis labels 1, 2, 3, 4, 5 representing S:E 1:2, 1:10, 1:20, 1:40, 1:80, respectively.

Fig. R12. The orbital distributions of the rubrene and PYIT1 molecules.

Table R1. The calculated orbital energies of the rubrene and PYIT1 molecules.

Molecules	HOMO [eV]	LUMO [eV]	T ₁ [eV]	T ₂ [eV]	T ₃ [eV]	T ₄ [eV]	T ₅ [eV]	S ₁ [eV]	S ₂ [eV]
Rubrene	-4.62	-2.09	0.97	2.34	2.96	3.05	3.22	2.17	3.20
PYIT1	-5.35	-3.37	1.25	1.46	1.93	1.95	2.02	1.68	1.96

Comment 3. The UC efficiency is calculated by Eq. 5. Eq. 5 includes absorbance. The absorbance value of the film should be provided. Normally, the UC efficiency does not include absorbance, but EQE includes absorbance. Please clearly explain how to calculate the efficiency values.

Author reply: Thanks for the reviewer's suggestions. Many works have proven that EQE measurements of TTA-UC devices achieved using an integrating sphere are affected by factors such as reabsorption and some other coupling losses. [*Chem. Soc. Rev.* **2020**, *49*, 6529] Therefore, relative methods are widely used to measure the performance of TTA-UC devices. According to the literature, the EQE of the UC emission can be calculated by the following **Eq. R2**:

$$EQE = \left(\frac{I_{UC}}{I_{std}}\right) \left(\frac{P_{std}}{P_{UC}}\right) (1 - 10^{-A_{std}}) QE_{std} \quad (\mathbf{R2})$$

Baldo et al. mentioned in [*Adv. Mater.* **2020**, *32*, 1908175.] that the UC efficiency (Φ_{UC}) is defined as the ratio of the number of upconverted photons emitted to the number of low-energy photons absorbed, which can be further expressed by **Eq. R3**. The maximum Φ_{UC} is 50%.

$$\Phi_{UC} = EQE / (1 - 10^{-A_{UC}}) \quad (\mathbf{R3})$$

Then, the Φ_{UC} can be calculated through **Eq. R4**.

$$\Phi_{UC} = \left(\frac{I_{UC}}{I_{std}}\right) \left(\frac{P_{std}}{P_{UC}}\right) \left(\frac{1 - 10^{-A_{std}}}{1 - 10^{-A_{UC}}}\right) QE_{std} \quad (\mathbf{R4})$$

Therefore, according to these equations, when calculating the Φ_{UC} of TTA-UC devices using relative methods, the absorption intensities of both the sample and the reference sample at the excitation wavelength are required. In the manuscript, we have also included the absorption spectra needed for the calculations. We appreciate your suggestions, and now we have added the detailed absorbance values of the film at the excitation wavelength to the corresponding figures.

The detailed changes can be seen in **Page 14** and as follows:

“The absolute PLQE of the rubrene:DBP film under an excitation wavelength of 488 nm, measured by the spectrometer system with the integration sphere is $48.98 \pm 2.02\%$. The EQE of the UC emission can be calculated by the following Eq. 5:

$$EQE = \left(\frac{I_{UC}}{I_{std}}\right) \left(\frac{P_{std}}{P_{UC}}\right) (1 - 10^{-A_{std}}) QE_{std} \quad (5)$$

where A is the absorbance at the excitation wavelength. I is the PL intensity and P is the number of photons irradiated from the excitation light source. Subscripts “UC” and “std” refer to the UC emission and the standard sample, respectively. The Φ_{UC} is defined as the ratio of the number of upconverted photons emitted to the number of low-energy photons absorbed, which can be further expressed by Eq. 6.²⁴

$$\Phi_{UC} = EQE / (1 - 10^{-A_{UC}}) \quad (6)$$

Then, the Φ_{UC} can be calculated through Eq. 7.

$$\Phi_{UC} = \left(\frac{I_{UC}}{I_{std}}\right) \left(\frac{P_{std}}{P_{UC}}\right) \left(\frac{1 - 10^{-A_{std}}}{1 - 10^{-A_{UC}}}\right) QE_{std} \quad (7)$$

Fig. S5 TTA-UC emission and absorption spectra. The PL spectra of (a) PYIT1:PBQx-TCl:rubrene- and (b) PYIT1:PBQx-TCl:rubrene:DBP-based TTA-UC devices with various

excitation intensities. The absorption spectra of (c) PYIT1:PBQx-TCl:rubrene- and (d) PYIT1:PBQx-TCl:rubrene:DBP-based TTA-UC devices.

Comment 4. The authors mention “the emitter rubrene crystal” in line 75, and “ordered aggregation in solution state” in line 199. The physical properties of crystal and amorphous rubrene are totally different such as HOMO level and emission spectrum, and I think rubrene in this mixed film is amorphous. Please clearly explain the crystallinity of rubrene.

Author reply: Thanks for your comments and suggestions. In this work, we thoroughly investigated the rubrene crystallinity properties in neat and blend films through GIWAXS and in-situ absorption spectroscopy. For the rubrene neat film, we prepared it by spin-coating and then subjected it to thermal annealing at 100 °C. As shown in **Fig. R13a**, the neat rubrene film exhibits sharp scattering peaks accompanied by many diffraction spots, which indicates rubrene forms randomly oriented crystals. [*Materials* **2021**, *14*, 7247.] In the blend film, rubrene only shows broad peaks at $q \approx 0.68 \text{ \AA}^{-1}$ and 1.30 \AA^{-1} with isotropy, confirming a significant reduction in the crystallinity of rubrene. This is consistent with the transition of rubrene from a crystalline state to an amorphous state as reported in some literature (**Fig. R13b-c**). Therefore, describing rubrene as amorphous in the blend film is indeed more accurate. This also better matches what we aim to convey in our manuscript about the uniform dispersion of rubrene in PBQx-TCl and PYIT1. Following your suggestion, we have provided detailed explanations of the crystalline state of rubrene in both neat and blend films. The detailed modifications are shown in **Page 8** of the revised manuscript and are as follows:

“The phase transition processes of rubrene in the blend system exhibit similar trends with that of the neat system (Fig. S10d). However, there is an obvious blue shift in the absorption peak, which suggests a significant decrease in crystallinity of rubrene in the blend film.”

“Additionally, rubrene in the blend film only shows broad peaks at $q \approx 0.68 \text{ \AA}^{-1}$ and $q \approx 1.30 \text{ \AA}^{-1}$ with isotropy (Fig. 3h), also confirming a significant reduction in the crystallinity of rubrene. This is consistent with the transition of rubrene from a crystalline state to an amorphous state as reported in literature.⁵⁷ Therefore, rubrene exists in an amorphous state within the blend film and is uniformly distributed within the fibrous network formed by PYIT1 and PBQx-TCl.”

[Redacted]

Fig. R13. GIWAXS patterns of rubrene films from (a) our work, (b) *ACS Nano* **2023**, *17*, 22642; and (c) *Materials* **2021**, *14*, 7247.

Comment 5. Please add the information on DBP concentration in the main manuscript.

Author reply: Thanks for the reviewer's suggestions. Due to the energetics of the triplet state, which is approximately half the energy of the singlet state, rubrene can engage in both TTA and singlet fission (SF). Consequently, DBP is often introduced into the rubrene layer to efficiently capture singlet states from rubrene, thereby suppressing SF and enhancing the luminous efficiency of the corresponding devices. According to previous reports, the weight ratio of DBP to rubrene typically ranges from 0.5% to 1%. [*J. Phys. Chem. C* **2020**, *124*, 18132.] In this study, we used a ratio of 0.75 wt%. Following your suggestion, we have included the purpose and concentration of DBP used in the manuscript. The detailed modifications are shown in **Page 6** of the revised manuscript and are as follows:

“Due to the energetics of the triplet state, which is approximately half the energy of the singlet state, rubrene can engage in both TTA and singlet fission (SF). Consequently, tetraphenyldibenzoperiflanthene (DBP) is often introduced into the rubrene layer to efficiently capture singlet states from rubrene, thereby suppressing SF and enhancing the corresponding device performance.⁴⁸ The molecular structure, absorption and PL spectra of DBP are shown in Fig. S7. Here, we introduce DBP into the PYIT1:PBQx-TCl:rubrene system as the emitter with a

weight ratio relative to rubrene of 0.75 wt% and fabricated the PYIT1:PBQx-TCl:rubrene:DBP UC device using the same one-step method.”

Comment 6. The authors mention that PYIT1 is a donor in lines 76 and 94. But, I think PYIT1 is an acceptor. If the author intended to say PYIT1 is an energy donor for UC, it should be explained as a sensitizer because a donor in BHJ blend normally means electron donor.

Author reply: Thanks for the reviewer’s comments and suggestions. The TTA-UC system consists of two components: the sensitizer and the annihilator. In previous TTA-UC systems, the sensitizer is typically a single component responsible for absorbing photons to provide energy. In this work, we propose and utilise a BHJ-type (PYIT1:PBQx-TCl) sensitizer. In this BHJ, due to the optical bandgap, NIR photons (800 nm) can only excite PYIT1. Therefore, in this PYIT1:PBQx-TCl-based system, PYIT1 serves as the component providing energy, while PBQx-TCl is mainly used to provide the driving force for efficient charge separation. In the manuscript, for clarity, we simply refer to PYIT1 as the donor and PBQx-TCl as the acceptor. In fact, our intention in the manuscript is to convey that PYIT1 acts as the energy donor, while PBQx-TCl acts as the energy acceptor. We fully agree with the reviewer's point that in organic semiconductor systems, particularly in OPV systems, the component providing electrons is referred to as the donor, while the component accepting electrons is referred to as the acceptor. Therefore, directly referring to PYIT1 as the donor in the manuscript may indeed cause some confusion among readers. Hence, following the reviewer’s suggestion, we have made revisions in the manuscript to uniformly refer to PYIT1 as the energy donor within the sensitizer. The detailed modifications are shown in **Page 2** of the revised manuscript and as follows:

“The energy donor in sensitizer PYIT1 can efficiently absorb NIR photons to generate singlet excitons, which then separate to free charges at the interface between PYIT1 and PBQx-TCl (energy acceptor in sensitizer).”

Reviewer #3 (Remarks to the Author):

In this work, a high-performance organic bulk heterojunction (BHJ) is doped with rubrene and its corresponding singlet emitter molecule DBP to make solution processed films that show photon up-conversion based on triplet-triplet annihilation.

Author reply: We sincerely thank for your suggestions that led us to further improve the manuscript. We have revised our manuscript in accordance with your comments, as summarized below. Changes in the manuscript are highlighted in red.

Comment 1. The authors report a very high up-conversion quantum yield (QY), especially for thin film, though these QY measurements were not performed in an integrating sphere (this is not standard procedure for thin films).

Author reply: Thanks for the reviewer's comments. Estimation of the UC efficiency (Φ_{UC}) of TTA-UC system is essential to assess the performance of UC systems for practical applications. There are two commonly used methods to measure the Φ_{UC} . The first is an absolute method using an integrating sphere and spectrometer, which is broadly inspired by the work of de Mello et al. [*Adv. Mater.* **1997**, *9*, 230] The second commonly used method for Φ_{UC} measurement is a relative method which uses either a fluorimeter or spectrometer to compare the integrated UC emission to the integrated PL of an established standard sample. It has been widely proven that absolute method is considered to give an underestimate of Φ_{UC} due to the outcoupling losses, such as the reabsorption, which exhibit a large absorbance at the UC emission wavelength during multiple scattering. Therefore, for almost all TTA-UC systems, relative methods are commonly employed. [*Chem. Rev.* **2015**, *115*, 395]

Here, according to the literature, we roughly estimated the reabsorption effect by comparing the UC emission spectra of the PYIT1:PBQx-TCl:rubrene:DBP-based TTA-UC system with and without the integrating sphere. [*J. Phys. Chem. A* **2019**, *123*, 10197] The schematic diagram is shown in **Fig. R14a**. In Experiment A, the sample was placed inside the integrating sphere, and the laser beam was directed onto the UC film. Experiment B was similar to Experiment A except that the integrating sphere was removed without changing other configurations. The UC emission spectrum obtained in Experiment A represents the spectrum after reabsorption, while the emission spectrum outside the sphere in Experiment B is the UC emission with minimum reabsorption.

Compared with the photons obtained from emission spectra A and B, the effect of reabsorption is particularly severe for our TTA-UC device, resulting in nearly an order of magnitude outcoupling loss. We briefly analysed the reasons for this loss. We employed a dual-component sensitizer (PYIT1:PBQx-TCl), and the broad absorption of PYIT1 and PBQx-TCl significantly overlaps with the UC emission, as shown in **Fig. R14b**. Additionally, both PYIT1 and PBQx-TCl have large absorption coefficients of 1.9×10^5 and $1.0 \times 10^5 \text{ M}^{-1} \text{ cm}^{-1}$, respectively (**Fig. R14d-g**). Furthermore, the ratio of sensitizer to annihilator is relatively high (weight ratio of 1:2.5), and the thickness of the UC film is over 100 nm, which results in strong reabsorption of sensitizer for UC emission. Factors such as light scattering also contribute to other coupling losses.

Although the relative method may introduce some uncertainty, this work demonstrates the effectiveness of using donor/acceptor sensitizers for fabricating high-performance solid-state TTA-UC devices by comparing the performance of the TTA-UC devices we prepared. We have removed the performance comparison of devices from different studies and added **Fig. R14** as the new **Fig. S4**. The detailed explanations are shown in **Page 5** of the revised manuscript and as follows:

“Typically, an integrating sphere is usually used for measuring the absolute photoluminescence quantum efficiency (PLQE) of a material. However, for TTA-UC devices, direct PLQE measured using an integrating sphere can be affected by reabsorption of UC photons by sensitizers.⁷ We estimated the reabsorption effect by comparing the UC emission spectra of the PYIT1:PBQx-TCl:rubrene:DBP-based TTA-UC system with (A) and without (B) the integrating sphere (Fig. S4).⁴⁷ The UC emission spectrum obtained in Experiment A represents the spectrum after reabsorption, while the emission spectrum outside the sphere in Experiment B can be considered as the UC emission with minimum reabsorption effect. Comparing the photons obtained from emission spectra A and B, the effect of reabsorption is particularly severe for our TTA-UC device, resulting in nearly an order of magnitude outcoupling loss. This is due to the significant overlap between our UC emission and the absorption spectra of the D-A sensitizer. Additionally, as shown in Fig. S4d-g, both PYIT1 and PBQx-TCl exhibit relatively large absorption coefficients, at 1.9×10^5 and $1.0 \times 10^5 \text{ M}^{-1} \text{ cm}^{-1}$, respectively. Besides, experimental output losses such as

waveguiding, scattering, and inner-filter effects can also induce outcoupling losses. Therefore, a relative method was used to estimate the device Φ_{UC} .”

Fig. R14. **a-b** Experimental configurations for collecting UC spectra A and B. **c** The normalised absorption and UC emission spectra of the PYIT1:PBQx-TCl:rubrene:DBP film. **d-g** Absorption spectra of PYIT1 and PBQx-TCl with different concentrations in solutions and the fitted absorption coefficients.

Comment 2. The best performing films contain both DBP and rubrene, even though the solar cell power conversion efficiency (PCE) drops from 31.7% to 5.2% with the addition of rubrene. While the physical and electrical characterization in this paper seem to be complete, e.g. beautiful atomic force microscopy, XRD and PCE curves, this paper makes a strong claim that charge transfer states are involved. In particular, the authors claim that triplet excitons are created from hole transfer from the acceptor to rubrene. There is little evidence for this hypothesis. Yes, there might be charge transfer states in the 17% PCE organic solar cell, but the morphology is different once DBP and rubrene are added, as seen by the decrease in PCE. If there is hole transfer, then where is the electron transfer occurring from, in order to create the triplet exciton on rubrene?

Author reply: Thanks for the reviewer's comments. As devices capable of converting photons into charge carriers, OPV cells provide a reliable means of assessing charge generation and recombination properties in the corresponding BHJs. Extensive research has shown that the working process of OPV cells is roughly as follows: Firstly, photoexcitation of the donor or acceptor generates an exciton. Subsequently, the exciton diffuses to the donor-acceptor interface and dissociates into free charges after undergoing charge transfer (CT) states. Following this, these charge carriers are transported and collected at the respective electrodes, as depicted in **Fig. R15a**. [*Nat. Rev. Mater.* **2018**, *3*, 18003] In which, the CT states play a crucial role in exciton dissociation and charge recombination processes (**Fig. R15b**). [*Nat. Rev. Mater.* **2019**, *4*, 689]

[Redacted]

Fig. 15. a The working mechanism of a BHJ-type OPV cell [*Nat. Rev. Mater.* **2018**, *3*, 18003]. **b** Electronic-state diagram illustrating the processes involved in photoinduced charge-carrier formation in an OPV cell [*Nat. Rev. Mater.* **2019**, *4*, 689].

In our work, OPV cells were fabricated to demonstrate the excellence of the PYIT1:PBQx-TCl-based BHJ. Under 1-sun illumination, the PYIT1:PBQx-TCl-based cell exhibits a PCE exceeding 17%, coupled with a high V_{OC} of 0.925 V. Additionally, under 808 nm laser illumination at 10 mW cm^{-2} , the PCE is 31.7%. In the manuscript, we mentioned that the energy of the charge transfer state (E_{CT}) obtained through fitting the high-sensitivity EQE (HEQE) and EL spectra is 1.47 eV (**Fig. S18a**). In previous works, transient absorption (TA) spectroscopy allowed clear observation of photoinduced absorption (PIA) signals corresponding to localized excitons (LE) and CT excitons in similar BHJs. [*J. Am. Chem. Soc.* **2020**, *142*, 12751] We also further collected TA signals of PYIT1:PBQx-TCl film in the NIR region with 800 nm excitation,

and the results are shown in **Fig. R16a** and **c**. According to the report, we observed distinct absorption of the PYIT1 LE at 850 nm, and at 940 nm, absorption corresponding to the CT exciton between PYIT1 and PBQx-TCl was observed [*J. Am. Chem. Soc.* **2021**, *143*, 7599]. The kinetics curves also exhibit characteristics consistent with those of LE and CT excitons (**Fig. R16e**).

Fig. R16. The TA images of (a) PYIT1:PBQx-TCl and (b) PYIT1:PBQx-TCl:rubrene films. TA spectra of (c) PYIT1:PBQx-TCl and (d) PYIT1:PBQx-TCl:rubrene films at different delay times. The normalised decay curves probed at various wavelengths recorded from (e) PYIT1:PBQx-TCl and (f) PYIT1:PBQx-TCl:rubrene films.

After adding rubrene, under 1-sun illumination, the PCE of the PYIT1:PBQx-TCl:rubrene-based cell decreased to 5.21% (under 808 nm laser illumination at 10 mW cm^{-2} , the PCE is 16.36%). It should be noted that compared to the PYIT1:PBQx-TCl-based cell, the device parameter of the PYIT1:PBQx-TCl:rubrene-based cell demonstrates a slight decrease in V_{OC} , from 0.92 V to 0.90 V, while the notable decreases were observed in J_{SC} (from 23.89 to 12.09 mA cm^{-2}) and FF (from 75.84% to 47.75%). As mentioned by the reviewer, the introduction of rubrene indeed changes the morphology, which we also observed through the detailed morphological characterizations. Combining the changes in device parameters, we can initially conclude that in the PYIT1:PBQx-TCl:rubrene system, under 800 nm excitation, the excitons were effectively separated at the interface of PYIT1 and PBQx-TCl. However, during the charge transport and

collection processes, significant non-geminate recombination takes place due to the influence of morphological changes, leading to a pronounced decrease in J_{SC} and FF.

More importantly, we can prove that the charge transfer process in the PYIT1:PBQx-TCl:rubrene system is similar to that in the PYIT1:PBQx-TCl system with additional evidences. **Firstly**, in the TA image of PYIT1:PBQx-TCl:rubrene system in the visible region following 800 nm excitation, we observed distinct GSB signals from PBQx-TCl, while no corresponding GSB signal from rubrene was observed (**Fig. 4a**). This observation is similar to what was observed in the PYIT1:PBQx-TCl system (**Fig. S14c**). **Secondly**, the E_{CT} obtained through fitting the HEQE and EL spectra was also highly consistent with that obtained in the PYIT1:PBQx-TCl system (1.47 eV) (**Fig. S18b**). **Furthermore**, in the TA image of PYIT1:PBQx-TCl:rubrene system in the NIR region following 800 nm excitation, we observed absorption signals corresponding to the LE of PYIT1 and the CT exciton between PYIT1 and PBQx-TCl at the same positions as those of the PYIT1:PBQx-TCl system (**Fig. R16b, d and f**). These results suggest that under 800 nm excitation, following the generation of excitons by PYIT1, holes are initially transferred to PBQx-TCl, forming CT states at the interface of PYIT1 and PBQx-TCl. For electrons, due to energy level constraints, they can only reside in the LUMO of PYIT1. The holes further transfer to rubrene via PBQx-TCl, leading to recombination at the interface between PYIT1 and rubrene. We have added some additional explanations in the manuscript to describe the charge transfer processes. We also added **Fig. R16** as the new **Fig. S16**. The detailed modifications are shown in **Page 10** of the revised manuscript and as follows:

“For electrons, due to energy level constraints, they can only remain in the LUMO of PYIT1.”

“The TA spectra in the NIR region with 800 nm excitation were also measured to demonstrate the generation of CT states between PYIT1 and PBQx-TCl, and the results are shown in Fig. S16. For PYIT1:PBQx-TCl, we observed distinct absorption of the PYIT1 localized exciton (LE) at 850 nm, and at 940 nm, absorption corresponding to the CT exciton between PYIT1 and PBQx-TCl is also observed.⁶¹ The decay kinetics of 850 nm and 940 nm signals also exhibit characteristics consistent with those of LE and CT excitons. For the PYIT1:PBQx-TCl:rubrene system, we observed absorption signals corresponding to the LE of PYIT1 and the CT exciton between PYIT1 and PBQx-TCl at the similar positions. Additionally, in both of the PYIT1:PBQx-TCl and

PYIT1:PBQx-TCl:rubrene TA spectra in visible region, we also extracted absorption signals at 615 nm. According to previous report, these signals correspond to charge-separated (CS) states between PYIT1 and PBQx-TCl.⁶¹ The decay dynamics of the CS states are shown in Fig. S17. The results further confirm our description that photogenerated excitons initially separated at the PYIT1:PBQx-TCl interface, followed by subsequent TTA-UC processes.”

Comment 3. If the triplet exciton was created on rubrene, then a clear triplet excited state absorption should be observed. This is not seen in any of the spectra, whether in the main paper or SI.

Author reply: Thanks for the reviewer’s very valuable comments. The generation of triplet states is a prerequisite for TTA-UC and is also an important factor in proving that the UC mechanism is TTA. For triplet states, TA spectroscopy is a very common technique used to capture this signal. As shown in **Fig. R17**, through TA spectroscopy, researchers have observed the PIA signal of rubrene’s T_1 to T_n transition around 510 nm.

[Redacted]

Fig. R17. TA spectra of rubrene films at different delay times from previous works of (a) *J. Mater. Chem. C* **2022**, *10*, 4684, (b) *J. Mater. Chem. C*, **2021**, *9*, 4359 and (c) *JACS Au* **2021**, *1*, 2188. Red lines indicate the triplet excited state absorption signals.

In fact, in the TA spectra obtained from our experiments, we also observed such similar PIA signals around 510 nm in the PYIT1:PBQx-TCl:rubrene system. In contrast, such signal was not observed in the PYIT1:PBQx-TCl system (**Fig. R18a-b**). Furthermore, this signal remains pronounced after 7 ns, consistent with the long lifetime of triplet states (**Fig. R18c**). Therefore, we reasonably attribute the PIA around 510 nm in the PYIT1:PBQx-TCl:rubrene system to the $T_1 \rightarrow T_n$ transitions in rubrene. However, due to the presence of GSB signals from PYIT1 at this wavelength, the absorption signal of rubrene’s triplet states is relatively weak. To further validate

this, we performed additional experiments, which obtained very similar results, as shown in **Fig. R19**. This further confirms the generation of rubrene triplet states.

Fig. R18. TA spectra of (a) PYIT1:PBQx-TCl and (b) PYIT1:PBQx-TCl:rubrene films at different delay times. (c) The normalised decay dynamics probed at 510 nm recorded from PYIT1:PBQx-TCl and PYIT1:PBQx-TCl:rubrene films.

Fig. R19. The TA images of (a) PYIT1:PBQx-TCl and (b) PYIT1:PBQx-TCl:rubrene films with 800 nm excitation. TA spectra of (c/e) PYIT1:PBQx-TCl and (d/f) PYIT1:PBQx-TCl:rubrene films at different delay times.

In order to provide clearer explanations, we have added relevant information to the revised manuscript. We also added **Fig. R18** as the new **Fig. S15**. The detailed modifications are shown in **Page 10** of the revised manuscript and as follows:

“Furthermore, we also observe a photoinduced absorption (PIA) signal around 510 nm in the PYIT1:PBQx-TCl:rubrene system, as shown in Fig. S15. In contrast, such signal was not observed

in the PYIT1:PBQx-TCl system. Similar PIA features in the same spectral region have been explicitly assigned to $T_1 \rightarrow T_n$ transitions in rubrene.⁶⁰ However, it is difficult to accurately extract this absorption signal separately due to overlap with the GSB signals of PYIT1. This signal remains pronounced after 7 ns, consistent with the long lifetime of triplet states. It confirms the generation of rubrene triplet states. The generation of the triplet states is then followed by the fusion of the triplet states of two rubrene molecules into a higher-lying, emissive singlet exciton, thus completing the UC emission process.”

Comment 4. There are many papers showing that rubrene can be directly excited with the 800 nm laser, e.g. by Chris Bardeen. Did the authors do control experiments showing that their excitation density for transient absorption is below this threshold for the two-photon excitation of rubrene?

Author reply: Thanks for the reviewer’s comments. We fully understand and agree with your concerns. Two-photon absorption (TPA) represents a significant pathway for photon UC. As you mentioned, Bardeen et al. demonstrated an alternate pathway to NIR UC in neat rubrene crystals: resonantly enhanced two-photon absorption via a weakly allowed interband state. They demonstrated TPA of rubrene under 800 nm excitation, achieving bright luminescence, as depicted in **Fig. R20a**. [*J. Phys. Chem. C* **2018**, *122*, 17632] Similarly, Karki et al. demonstrated that the triplet states of rubrene can be directly excited from the ground state using 800 nm excitation, followed by triplet state fusion and achieving UC emission. (**Fig. R20b**). [*Phys. Rev. B* **2021**, *104*, L140308]

[Redacted]

Fig. R20. The examples of TPA in rubrene thin films from previous works of (a) *J. Phys. Chem. C* **2018**, *122*, 17632 and (b) *Phys. Rev. B* **2021**, *104*, L140308.

In our study, we also observed TPA of rubrene when the excitation intensity was sufficiently high. Additionally, PBQx-TCl also can exhibit TPA features under higher intensities, as shown in

Fig. R21a-b. Therefore, we employed relatively lower excitation intensities ($< 10 \mu\text{J cm}^{-2}$) in our experiments. Under such intensities, TPA of rubrene thin films has not been observed, as illustrated in **Fig. R21c**, where, apart from some background signal, no rubrene-related emissions were observed.

Fig. R21. **a** The TA image and **(b)** TA spectra of PBQx-TCI neat film under high excitation power at 800 nm. **c** The TA image of rubrene neat film under 800 nm excitation ($10 \mu\text{J cm}^{-2}$).

We have added some descriptions to **Page 9** of the revised manuscript. The detailed modifications are as follows:

“For TA measurements, employing high excitation energies may induce two-photon absorption (TPA) effect in rubrene. To avoid this phenomenon, we used excitation energies smaller than $10 \mu\text{J cm}^{-2}$. As shown in Fig. S13, under this excitation energy, rubrene does not exhibit TPA effect.

Fig. S13 Transient absorption (TA) spectroscopic study. The TA image of rubrene neat film under 800 nm excitation.”

REVIEWER COMMENTS

Reviewer #1 (Remarks to the Author):

The authors have addressed my concerns and comments and greatly improved the paper in my opinion. I have no further concerns and recommend this work for publication.

Reviewer #2 (Remarks to the Author):

My concerns are addressed by the additional experiments and discussions.

Reviewer #3 (Remarks to the Author):

It appears that the authors are unable to address reviewer comments, specifically about the quantum yield measurements, and about evidence for the formation of the rubrene triplet state. With respect to the former, the authors glibly claim that "results obtained within different laboratories using the same relative method may also vary due to the sensitivity of measurements to the optical setups, making direct comparisons less rigorous". This implies that all the measurements are arbitrary and do not withstand scrutiny. This is not true. Internal and absolute quantum yield measurements have certain protocols to be followed for accurate measurements, e.g. for the internal quantum yield: the inner filter effect, correcting for reabsorption, for the standard having the same optical density at the excitation as the sample, etc. It does not sound like the authors have much experience performing quantum yield measurements. It is not clear if the upconversion quantum yield measurements are to be trusted.

Secondly, the transient absorption (TA) data cuts off at 5ns for the longest time point measured. As a result, the authors are unable to conclusively show data that resembles a rubrene triplet excited state absorption that they have kindly summarized from literature (Fig. R17). Since the triplet lives long, the authors should increase the time window for the TA spectroscopy, and also use thinner films. The initial ground state bleach is huge- it means light is unable to penetrate the sample

Response Letter to Reviewers' Comments

Reviewer #1 (Remarks to the Author):

The authors have addressed my concerns and comments and greatly improved the paper in my opinion. I have no further concerns and recommend this work for publication.

Author reply: We appreciate to the reviewer for supporting this research work to be published in *Nature Communications*.

Reviewer #2 (Remarks to the Author):

My concerns are addressed by the additional experiments and discussions.

Author reply: We appreciate to the reviewer for supporting this research work to be published in *Nature Communications*.

Reviewer #3 (Remarks to the Author):

Comment 1. It appears that the authors are unable to address reviewer comments, specifically about the quantum yield measurements, and about evidence for the formation of the rubrene triplet state. With respect to the former, the authors glibly claim that "results obtained within different laboratories using the same relative method may also vary due to the sensitivity of measurements to the optical setups, making direct comparisons less rigorous". This implies that all the measurements are arbitrary and do not withstand scrutiny. This is not true.

Author reply: We thank the reviewer for the comment. As mentioned by the *Reviewer 1* in the last response letter, there is some controversy regarding the comparison of up-conversion efficiencies (Φ_{UCS}) measured by different methods. We recognise this issue and have replied comprehensively that other testing conditions and equipment may also slightly influence the results. Therefore, to ensure the rigour of our manuscript, we have removed the comparison of Φ_{UCS} of different TTA-UC reports from the original manuscript, to avoid unfair comparison arising from different methods. However, our intention was to indicate that directly comparing

results from different methods is inappropriate, not to imply that the measurements are arbitrary and do not withstand scrutiny.

Comment 2. Internal and absolute quantum yield measurements have certain protocols to be followed for accurate measurements, e.g. for the internal quantum yield: the inner filter effect, correcting for reabsorption, for the standard having the same optical density at the excitation as the sample, etc. It does not sound like the authors have much experience performing quantum yield measurements. It is not clear if the up-conversion quantum yield measurements are to be trusted.

Author reply: We thank the reviewer for the comment and understand the reviewer's concern. As we comprehensively discussed in the last response letter, for the TTA-UC systems, due to reabsorption and other outcoupling losses, the Φ_{UC} is typically measured using relative method. We also conducted extensive literature research and found that almost all reported TTA-UC system Φ_{UC} measurements were conducted using relative methods. [*Chem. Rev.* **2015**, *115*, 395] Moreover, relevant reports have widely proven that the relative methods are accurate and without any controversy. In our work, we demonstrated through experiments that our TTA-UC devices exhibit strong reabsorption effect; hence, we also used a relative method for Φ_{UC} measurement. We pay very close attention to the accuracy of the measurements. Here, the reliability of our test results can be demonstrated by the following points: (a) In terms of measurement methods, a most widely used relative method was used: $\Phi_{UC} = \left(\frac{I_{UC}}{I_{std}}\right) \left(\frac{P_{std}}{P_{UC}}\right) \left(\frac{1-10^{-A_{std}}}{1-10^{-A_{UC}}}\right) QE_{std}$; (b) The average PLQY values measured using absolute method for reference sample (Rubrene:DBP) are consistent with most literature reports; (c) The Φ_{UC} of the PYIT1:Rubrene-based UC device is well consistent with that of similarly reported UC devices. In addition, we measured other commonly used materials using this relative method, and the results were also comparable to reported values. These results indicate that our measured Φ_{UC} values are very reliable.

Comment 3. Secondly, the transient absorption (TA) data cuts off at 5 ns for the longest time point measured. As a result, the authors are unable to conclusively show data that resembles a rubrene triplet excited state absorption that they have kindly summarized from literature (Fig.

R17). Since the triplet lives long, the authors should increase the time window for the TA spectroscopy, and also use thinner films. The initial ground state bleach is huge- it means light is unable to penetrate the sample.

Author reply: We appreciate the reviewer's suggestion. The TA data shown in the manuscript cut off at ~8 ns. Therefore, it may not be very convincing to attribute the PIA signal around 510 nm to the triplet state of rubrene. The narrow time window in the TA setup is due to the constraints of the delay stage, which is a common issue in the femtosecond-nanosecond (fs-ns) TA setups. A suitable setup that excites the sample at ~800 nm, while allows for both ultrafast and long-time timescale TA measurements, can be rare. Besides, the reviewer mentioned that the initial ground state bleach (GSB) signal is very strong, questioning that the sample might be too thick and preventing light from adequately penetrating the sample. In fact, the samples used for our TA measurements have been carefully prepared to ensure relatively good transparency, and the thickness is around 60-70 nm. The observed strong initial GSB may be due to the high extinction coefficients of the components in the films.

We agree with the reviewer's statement and recognise the importance of capturing the full dynamics of the PIA signal around 510 nm, which can allow us to further provide more robust evidence for the formation of the rubrene triplet state. Therefore, TA spectroscopy was further carried out using an EOS nano- and microsecond (ns- μ s) spectrometer to measure the TA spectra over longer time windows. The measured results are shown in **Fig. R1**. The TA spectra exhibit long-lived PIA signals around 500 nm (**Fig. R1a** and **b**). These PIA peaks align well with the previously observed absorption peaks of the rubrene triplet state in the literature. Moreover, the PIA signal probed at 515 nm has a μ s-scale lifetime (**Fig. R1c**). These results confirm the formation of the rubrene triplet state in our TTA-UC system, further proving that the UC mechanism is based on the TTA process of rubrene triplet excitons. To illustrate this more clearly, we have added the ns- μ s TA spectra and corresponding explanations to the revised manuscript. The detailed modifications are shown in **Page 10 and 13** of the **revised manuscript** and **Page 10 (Figure S15) of the revised SI**, as follows:

Fig. R1. The PIA absorption of rubrene triplet states. **a** The ns- μ s TA spectral map of PYIT1:PBQx-TCl:rubrene film. **b** The ns- μ s TA spectra of PYIT1:PBQx-TCl:rubrene films at different delay times. **c** The normalised decay curves probed at 515 nm recorded from PYIT1:PBQx-TCl:rubrene films.

On Page 10 of main text:

“To further confirm that the PIA signals originate from the triplet state of rubrene, TA spectroscopy was further carried out using an EOS nano- and microsecond (ns- μ s) spectrometer. As shown in Fig. S15d-e, these PIA peaks around 510 nm align well with the previously observed absorption peaks of the rubrene triplet state.⁶⁰ Moreover, the PIA signal probed at 515 nm shows a μ s-scale lifetime (Fig. S15f). These results confirm the formation of the rubrene triplet states. The generation of the triplet states is then followed by the fusion of the triplet states of two rubrene molecules into a higher-lying, emissive singlet exciton, thus completing the UC emission process.”

On Page 13 of main text:

“The nano- and microsecond TA spectra were measured using a pump-probe nanosecond TA spectrometer with an extended time window (EOS, Ultrafast Systems).”

REVIEWERS' COMMENTS

Reviewer #3 (Remarks to the Author):

The authors have addressed my comments. Thank you.